# Inflammation causes insulin resistance in mice via interferon regulatory factor 3 (IRF3)-mediated reduction in FAHFA levels

Shuai Yan[1,2], Anna Santoro[1,2], Micah J. Niphakis[3], Antonio M. Pinto [4], Christopher L. Jacobs [1,2], Rasheed Ahmad [5], Radu M. Suciu[3], Bryan R. Fonslow[3], Rachel A. Herbst-Graham[3], Nhi Ngo[3], Cassandra L. Henry[3], Dylan M. Herbst[3], Alan Saghatelian [4], Barbara B. Kahn [1,2,6] & Evan D. Rosen [1,2,6] ✉

Obesity-induced inflammation causes metabolic dysfunction, but the mechanisms remain elusive. Here we show that the innate immune transcription factor interferon regulatory factor (IRF3) adversely affects glucose homeostasis through induction of the endogenous FAHFA hydrolase androgen induced gene 1 (AIG1) in adipocytes. Adipocyte-specific knockout of IRF3 protects male mice against high-fat diet-induced insulin resistance, whereas overexpression of IRF3 or AIG1 in adipocytes promotes insulin resistance on a high-fat diet. Furthermore, pharmacological inhibition of AIG1 reversed obesity-induced insulin resistance and restored glucose homeostasis in the setting of adipocyte IRF3 overexpression. We, therefore, identify the adipocyte IRF3/AIG1 axis as a crucial link between obesity-induced inflammation and insulin resistance and suggest an approach for limiting the metabolic dysfunction accompanying obesity.

In the setting of persistent caloric excess, pathological white adipose tissue (WAT) expansion is accompanied by a state of chronic inflammation, sometimes termed "meta-inflammation", leading to insulin resistance and, potentially, type 2 diabetes[1]. WAT inflammation is associated with recruitment[2,3] and proliferation[4] of macrophages and other immune cells, which secrete a variety of proinflammatory cytokines[5]. These cytokines, combined with other inflammatory stimuli that increase in obesity (e.g., LPS from the gut and cell-free DNA from dying adipocytes)[6,7], cause insulin resistance and related metabolic dysfunction.

Despite clear data demonstrating the role of inflammation in insulin resistance, the mechanism by which this occurs is still unclear. Most attention has focused on the acute effects of inflammation on proximal insulin signaling, despite evidence suggesting that the time course over which cytokine signals work (minutes to a few hours) is not concordant with the time course over which insulin action is affected (several hours to days)[8]. An alternative explanation is that inflammatory factors activate critical transcriptional pathways that alter the receptiveness of WAT and other tissues to insulin. This notion is bolstered by observations that several transcription factors (e.g., PPARγ) are known to alter insulin sensitivity, an effect shared by drugs or genetic manipulations that affect chromatin structure[9–11].

It is common to invoke increased activity of nuclear factor κB (NF-κB) in the setting of inflammation. NF-κB is sequestered in the cytosol bound to the inhibitor IκB protein until phosphorylation of IκB by IKKβ releases it to enter the nucleus[12]. Manipulations of IKKβ show

[1]Division of Endocrinology, Diabetes, and Metabolism, Beth Israel Deaconess Medical Center, 330 Brookline Ave, Boston, MA 02215, USA. [2]Harvard Medical School, 25 Shattuck St, Boston, MA 02130, USA. [3]Lundbeck La Jolla Research Center Inc., 10835 Road To The Cure Dr. #250, San Diego, CA 92121, USA. [4]The Salk Institute for Biological Studies, 10010 N. Torey Pines Rd, La Jolla, CA 92037-1002, USA. [5]Immunology and Microbiology Department, Dasman Diabetes Institute, Jasim Mohamad Al Bahar St., Kuwait City, Kuwait. [6]Broad Institute of Harvard and MIT, 320 Charles St., Cambridge, MA 02141, USA. ✉e-mail: erosen@bidmc.harvard.edu

the predicted effects on insulin action, especially in the liver[13]. This effect, however, is likely to be independent of NF-κB, given that (a) IKKβ also phosphorylates insulin signaling proteins directly[14], (b) overexpression of the active p65 subunit of NF-κB in adipocytes maintains insulin sensitivity in both age-related and diet-induced obesity in mice, despite proinflammatory gene expression in adipose tissue[15,16], and (c) mice lacking the regulatory p50 subunit have increased NF-κB activity and yet display resistance to diet-induced obesity and preserved insulin action[17,18]. Taken together, these studies indicate that transcriptional pathways other than NF-κB must mediate the effect of inflammation on insulin resistance. These observations also suggest that the classic proinflammatory cytokine portfolio induced by NF-κB and similar transcription factors may not be driving forces in insulin resistance, as is commonly assumed.

Another class of transcription factors with major roles in both innate and adaptive immunity are the interferon regulatory factors (IRFs), comprised of 9 members (IRF1–IRF9). IRFs have been implicated in a wide range of immune functions, including lymphopoiesis, macrophage differentiation, and antiviral defense[19]. We have previously determined the role of several IRFs in adipogenesis[20] and further demonstrated that IRF4 is an important regulator of adipocyte lipolysis[21] and thermogenesis[22]. The family member most associated with transducing proinflammatory signals is IRF3[23]. IRF3 expression is induced in the adipose tissue of people and rodents with obesity[24], and mice lacking IRF3 globally are resistant to diet-induced obesity, an effect recapitulated in mice lacking IRF3 specifically in adipocytes. Conversely, expression of a constitutively active form of IRF3 in adipocytes promotes weight gain[25]. This effect on adiposity is mediated by direct activation of *Isg15* gene expression and protein ISGylation in adipocytes, which results in reduced glycolysis and lactate production and strongly dampens adipose thermogenesis[25]. In addition to repressing thermogenesis, IRF3 in adipocytes may also exert weight- and ISGylation-independent effects on insulin sensitivity and glucose homeostasis, although this was not proven definitively, and no downstream mechanism has been identified[25]. This is in contrast to liver, where IRF3 was shown to impact hepatic insulin action through the direct induction of the protein phosphatase 2 A (PP2A) subunit *Ppp2r1b*[26].

Here we find that inflammatory stimuli modulate insulin action in adipocytes in a cell autonomous and IRF3-dependent manner. High-fat fed adipocyte-specific *Irf3* deficient (FI3KO) mice studied at thermoneutrality exhibit improved insulin and glucose tolerance, while adipocyte-specific overexpression of *Irf3* (FI3OE) promotes glucose intolerance and insulin resistance on HFD, also in a weight-independent manner. Mechanistically, we identify AIG1, an endogenous hydrolase of fatty acyl esters of hydroxy fatty acids (FAHFAs), as a downstream target of IRF3. Overexpression of AIG1 in adipose tissue impairs insulin and glucose tolerance in FI3KO mice. In addition, a novel small molecule inhibitor of AIG1 raises FAHFA levels and attenuates HFD-induced insulin resistance and glucose intolerance in FI3OE mice. These findings uncover a defined and direct mechanism by which inflammation impairs adipose insulin sensitivity and glucose homeostasis in obesity.

## Results

### Activation of toll-like receptors 3 and 4 provokes insulin resistance in cultured adipocytes in an IRF3-dependent manner

TLR4 activation has been shown to reduce insulin-stimulated glucose uptake in adipocytes[27]. We first sought to recapitulate this effect in cultured adipocytes derived from mouse stromal vascular fraction (SVF). Treatment with LPS for 2 days led to a dose-dependent decrease in insulin-stimulated glucose uptake (Fig. 1a). LPS treatment also increased phosphorylation of murine IRF3 at serine 388 (Fig. 1b), which is known to promote nuclear translocation, dimerization, and binding to DNA[28].

To determine whether IRF3 is required for LPS-induced insulin resistance, we knocked down IRF3 in mature adipocytes using a

doxycycline-inducible lentivirus. Knocking down IRF3 abolished the suppressive effect of LPS treatment on glucose uptake (Fig. 1c). We next isolated primary mouse inguinal SVF cells from *Irf3*flox/flox mice for in vitro differentiation into mature adipocytes, and then infected these cells with AAV-GFP (WT) or AAV-GFP-Cre (FI3KO). In mature adipocytes, IRF3 deletion led to a robust decrease in the expression of IRF3 target genes, including *Ccl5*, *Ifit1*, and *Isg15* (Supplementary Fig. 1a). Importantly, there was no significant difference in differentiation state between WT and FI3KO cells, as determined by the expression of mature adipocyte marker genes (Supplementary Fig. 1b). We observed higher p-AKT levels in FI3KO adipocytes compared with WT cells even in the absence of insulin, supporting the idea that endogenous adipocyte IRF3 represses basal Akt phosphorylation (Fig. 1d). Consistent with the signaling data, ablation of IRF3 enabled increased insulin-induced glucose uptake; a much smaller effect of IRF3 on glucose uptake was also observed in the absence of insulin (Fig. 1e). Finally, FI3KO adipocytes displayed greater sensitivity to the anti-lipolytic effects of insulin (Supplementary Fig. 1c).

To confirm these findings in human adipocytes, we used the Simpson-Golabi-Behmel syndrome (SGBS) preadipocyte cell strain[29]. As seen in mouse adipocytes, LPS treatment of human adipocytes led to a dose-dependent decrease in insulin-stimulated glucose uptake and an increase in IRF3 phosphorylation (Fig. 1f, g). Knockdown of IRF3 in SGBS cells abolished the suppressive effect of LPS treatment on insulin sensitivity, as measured by p-Akt and glucose uptake (Fig. 1h, i).

IRF3 can also be activated through TLR3 signaling [e.g., polyinosinic-polycytidylic acid (poly I:C) treatment][30]. We therefore determined the effect of the TLR3/IRF3 signaling axis in regulating glucose uptake in mouse SVF-derived adipocytes and human SGBS cells. Poly I:C treatment robustly decreased insulin-stimulated glucose uptake and increased phosphorylation of IRF3 in mouse SVF-derived adipocytes (Supplementary Fig. 1d, e). shRNA-mediated reduction of IRF3 fully abrogated poly I:C–induced insulin resistance in these cells (Supplementary Fig. 1f). Similar data were obtained from human adipocytes treated with poly I:C (Supplementary Fig. 1g-i).

### Constitutively active IRF3 induces insulin resistance in mouse and human adipocytes

We have shown that the double-mutant murine *Irf3* allele (S388D/S390D; hereafter designated as IRF3-2D) is constitutively active in vitro[25]. Concordant with the loss-of-function studies just described, overexpression of IRF3-2D in mouse SVF-derived adipocytes led to a significant decrease in insulin-stimulated glucose uptake and p-Akt in the absence of upstream activators (Fig. 2a). We also isolated mouse inguinal SVF cells from Irf3-2D mice, differentiated them into mature adipocytes, and infected these cells with AAV-GFP (WT) or AAV-GFP-Cre (FI3OE). IRF3 overexpression in mature adipocytes markedly increased the expression of target genes, without affecting adipogenesis (Supplementary Fig. 2a, b). FI3OE cells displayed decreased insulin-stimulated AKT phosphorylation and glucose uptake compared with WT cells; insulin was also less able to suppress lipolysis in these cells (Fig. 2b, c and Supplementary Fig. 2c). Human SGBS cells behaved similarly when IRF3-2D was expressed, showing reduced insulin-stimulated p-AKT and glucose uptake (Fig. 2d, e) in SGBS cells. Taken together with the loss-of-function studies, these data strongly suggest that adipocyte IRF3 is both necessary and sufficient to promote insulin resistance in adipocytes, and that it does so in a cell-autonomous and species-independent manner.

### Adipocyte-specific knockout of *Irf3* attenuates HFD-induced insulin resistance in mice at thermoneutrality

In our previous work, we demonstrated that fat-specific IRF3 knockout (FI3KO) mice have increased adipose thermogenesis and are protected from diet-induced obesity[25], which confounds the ability to determine whether changes in insulin sensitivity are dependent or independent

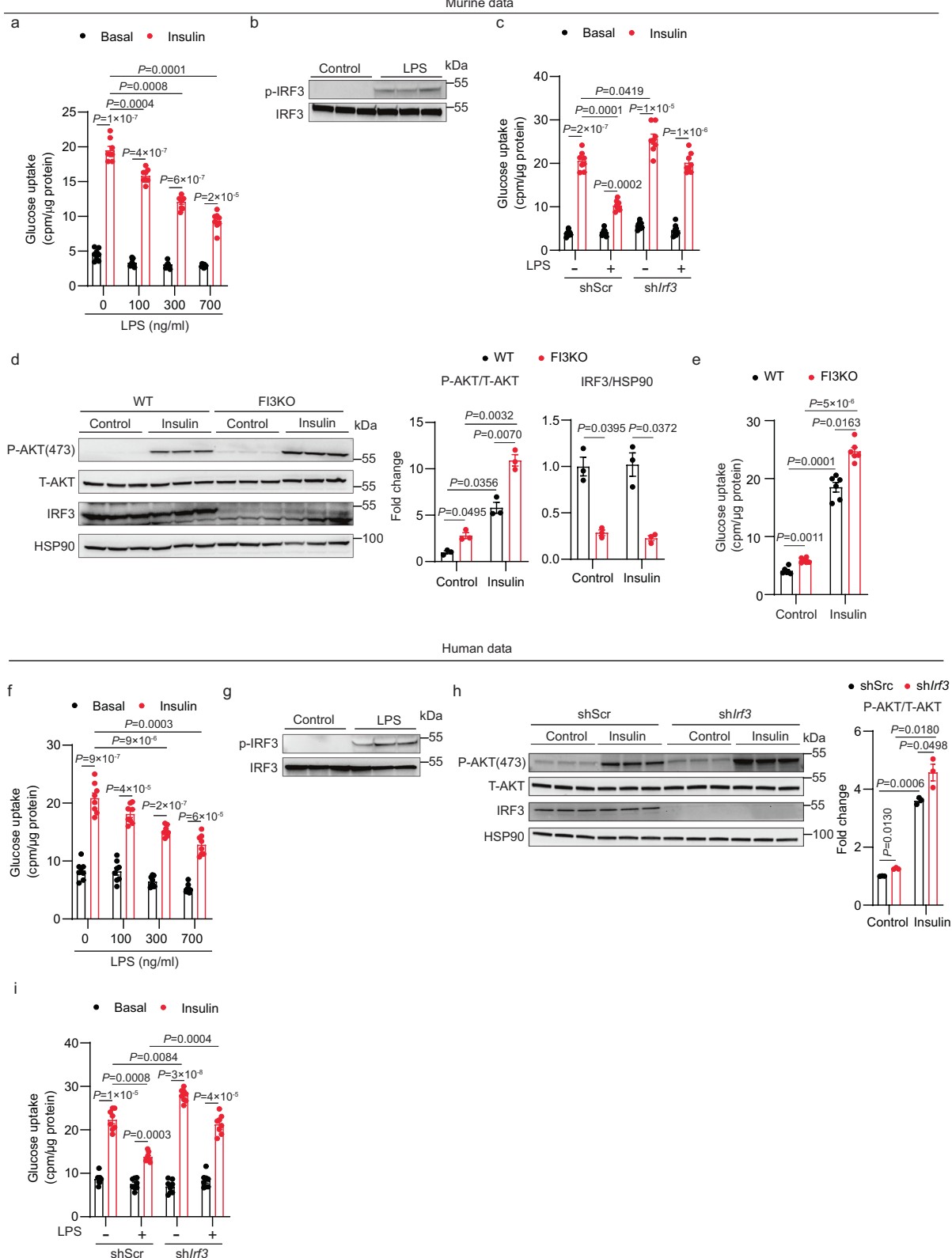

of changes in adiposity. To avoid this issue, we housed mice under thermoneutral (TN) conditions (30 °C) at which there is no requirement for adaptive thermogenesis to maintain body temperature (Supplementary Fig. 3a)[31]. After 16 weeks of HFD feeding, weight gain, adiposity, and food intake were not different between male FI3KO and WT mice (Supplementary Fig. 3b–d). However, despite the absence of protection from diet-induced obesity, FI3KO mice exhibited markedly

improved insulin sensitivity and glucose tolerance (Fig. 3a-c). Consistent with the improved glycemia, FI3KO mice exhibited significantly increased insulin-stimulated AKT phosphorylation within epididymal white adipose tissue (eWAT), inguinal WAT (iWAT), skeletal muscle, and liver, but not brown adipose tissue (BAT) (Fig. 3d). To understand the tissue-specific basis of the insulin sensitivity phenotype in FI3KO mice at a more quantitative level, we performed insulin-stimulated

**Fig. 1 | IRF3 mediates insulin resistance in response to TLR4 ligands in cultured adipocytes. a** Glucose uptake in mouse adipocytes after treatment with varying doses of LPS for 2 days ($n = 8$). **b** Western blot showing phosphorylation of murine IRF3 (S388) in mouse adipocytes after 30 min of LPS (700 ng/ml) treatment. **c** Glucose uptake in mouse adipocytes transduced with lentivirus expressing shRNA against *Irf3* or shScr control hairpin in the absence or presence of LPS (700 ng/ml) ($n = 8$). **d** Western blot of pAKT (S473) and IRF3, **e** Glucose uptake ($n = 6$) in WT and FI3KO SVF-derived adipocytes treated with control or 100 nM insulin. **f** Glucose uptake in human SGBS adipocytes after treatment with varying doses of LPS for 2 days ($n = 8$). **g** Western blot showing phosphorylation of human IRF3 (S396) in human SGBS adipocytes after 30 mins of LPS (700 ng/ml) treatment. **h** Western blot of pAKT (S473) and IRF3 in human SGBS adipocytes transduced with lentivirus expressing shRNA against *Irf3* or a scrambled control hairpin (shScr). **i** Glucose uptake in human SGBS adipocytes transduced with lentivirus expressing shRNA against *Irf3* or shScr control hairpin in the absence or presence of LPS (700 ng/ml) ($n = 8$). Statistical significance was assessed by *three-way* ANOVA (**c** and **i**) or *two-way* ANOVA (**a, d, e, f,** and **h**). Data are expressed as mean ± SEM.

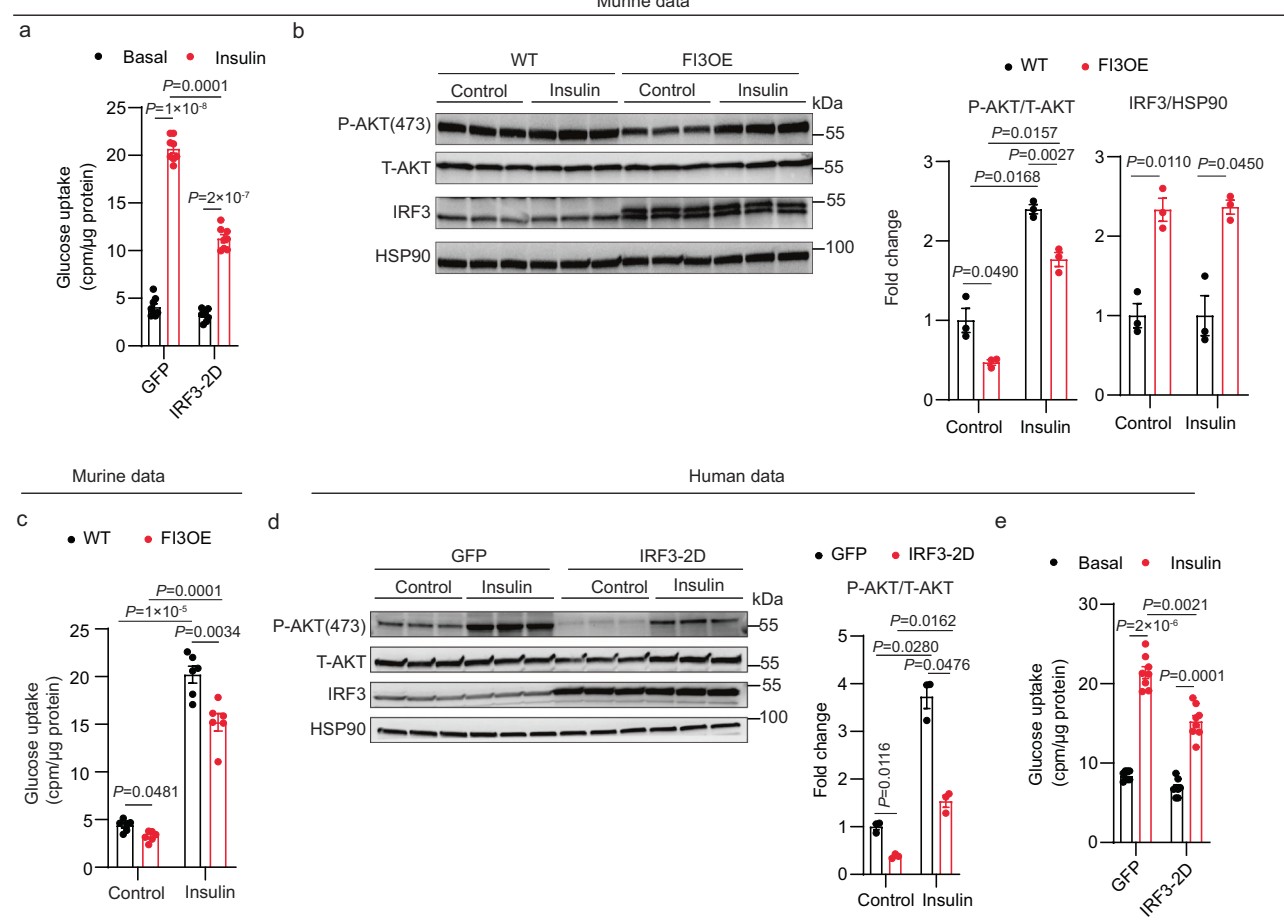

**Fig. 2 | IRF3 promotes insulin resistance in cultured mouse and human adipocytes. a** Glucose uptake in mouse adipocytes expressing GFP or constitutively active IRF3-2D mutant ($n = 8$). **b** Western blot of pAKT (S473) and IRF3 in WT and FI3OE SVF-derived adipocytes treated with control or 100 nM insulin. **c**, Glucose uptake ($n = 6$) in WT and FI3OE SVF-derived adipocytes treated with vehicle or 100 nM insulin. **d** Western blot of pAKT (S473) and IRF3 in human SGBS adipocytes transduced with lentivirus expressing GFP or IRF3-2D. **e** Glucose uptake in human SGBS adipocytes expressing GFP or IRF3-2D mutant ($n = 8$). Statistical significance was assessed by two-way ANOVA. Data are expressed as mean ± SEM.

glucose uptake of $^{14}$C-2-deoxyglucose in vivo in HFD-fed mice. Glucose uptake by BAT, iWAT, eWAT, quadriceps muscle, and heart was significantly increased in FI3KO mice compared to WT controls, indicating that the improved insulin sensitivity in FI3KO mice may result from greater glucose uptake into a wide range of insulin-sensitive tissues (Fig. 3e). Taken together, these results indicate that IRF3 deficiency in adipocytes is sufficient to attenuate obesity-induced insulin resistance at thermoneutrality.

**Adipocyte-specific overexpression of *Irf3* promotes HFD-induced insulin resistance in mice at thermoneutrality**

To further investigate the role of adipocyte IRF3 in insulin sensitivity, we used a line of knock-in mice (FI3OE) in which IRF3 is constitutively activated in adipocytes without the need for an external stimulus like LPS or poly I:C. FI3OE mice display enhanced weight gain when housed

at 23 °C[25]. After 16 weeks of HFD feeding at thermoneutrality, however, weight gain, adiposity, and food intake were comparable between FI3OE and WT mice (Supplementary Fig. 3e–g). Despite weighing the same as control mice, FI3OE mice exhibited markedly impaired insulin sensitivity and glucose tolerance (Fig. 4a–c). FI3OE mice exhibited a significant decrease in insulin-stimulated AKT phosphorylation in eWAT, BAT, iWAT, skeletal muscle, and liver (Fig. 4d). Consistent with this, glucose uptake by BAT, iWAT, eWAT, quadriceps muscle, and heart was significantly decreased in FI3OE mice compared to WT controls (Fig. 4e).

**IRF3 drives AIG1 expression, leading to reductions in intracellular FAHFA levels**

The finding that manipulating IRF3 levels in adipocytes affects insulin sensitivity in other tissues suggested that IRF3 might alter adipokine

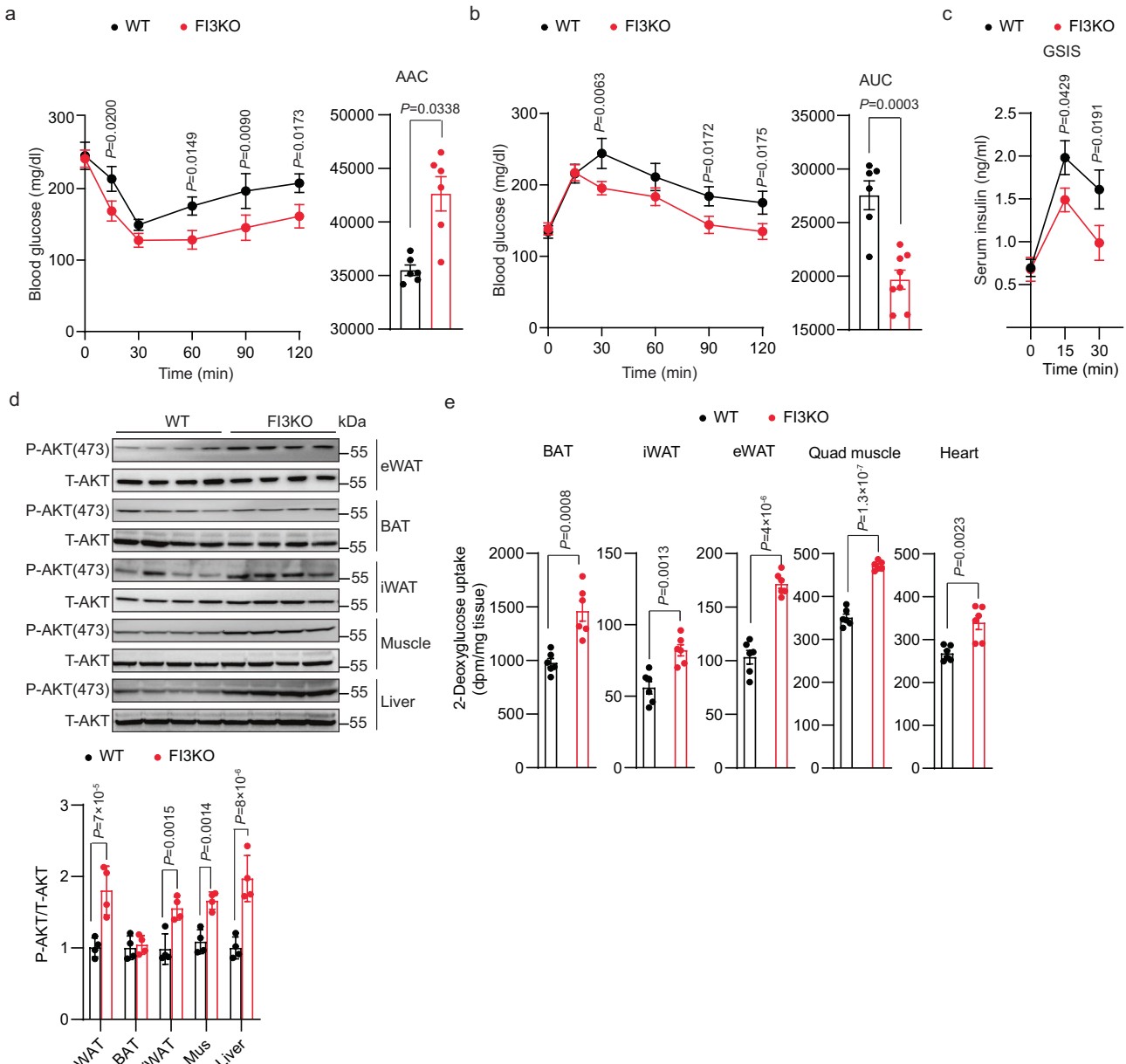

**Fig. 3 | Adipocyte-specific knockout of *Irf3* attenuates HFD-induced insulin resistance in mice at thermoneutrality.** Metabolic analysis of male WT and FI3KO mice ($n$ = 10) after 16 weeks of HFD feeding at thermoneutrality, including insulin tolerance test (**a**), glucose tolerance test (**b**), ad lib fed serum insulin levels (**c**), immunoblotting for P-AKT (S473) protein levels in eWAT, BAT, iWAT, muscle, and liver (**d**), in vivo 2-deoxyglucose uptake in BAT, iWAT, eWAT, quadriceps muscle and heart (**e**). Statistical significance was assessed by *two-way* ANOVA (**a**–**c**) or two-tailed Student's *t* test (**d**, **e**). Data in all panels are expressed as mean ± SEM.

secretion or action. To address this possibility, we co-cultured WT SVF-derived adipocytes with conditioned medium from either WT SVF-derived adipocytes (WT-CM), FI3KO SVF-derived adipocytes (FI3KO-CM), or FI3KO SVF-derived adipocytes treated with proteinase K (FI3KO-CM + PK). Surprisingly, both FI3KO-CM and FI3KO-CM + PK markedly increased p-AKT in WT adipocytes compared with WT-CM, indicating that a protease-insensitive substance(s) from FI3KO adipocytes can enhance insulin signaling (Fig. 5a).

IRF3 is a transcription factor, and so we sought to identify gene targets that could be credibly linked to changes in levels of insulin-sensitizing metabolites or lipids. To that end, we defined the transcriptional profiles of WT and FI3OE SVF-derived adipocytes (Fig. 5b and Supplementary Data 1). One significantly affected gene was androgen-induced gene 1 (*Aig1*) (Fig. 5c), which encodes a hydrolase that degrades members of the fatty acyl ester of hydroxy fatty acid (FAHFA) family of lipids produced by adipocytes, which have been

implicated as important endogenous insulin sensitizers[32–34]. qPCR and western blotting of SVF-derived adipocytes from FI3KO and FI3OE mice confirmed altered *Aig1* expression (Fig. 5d, e). Furthermore, *Aig1* was down-regulated in iWAT, eWAT, and BAT from FI3KO mice, and up-regulated in FI3OE mice (Fig. 5f-i). *Aig1* expression in other tissues, such as skeletal muscle, liver, and heart, was not altered in FI3KO and FI3OE mice (Supplementary Fig. 4a, b). Expression of another known FAHFA hydrolase, androgen-dependent TFPI-regulating protein (ADTRP), was unchanged in the tissues of FI3KO and FI3OE mice (Supplementary Fig. 4c, d).

We next used targeted lipidomics to quantify isomers from multiple FAHFA families, many of which are known to be important in insulin action and glucose homeostasis. We found that SVF-derived FI3KO adipocytes show elevated levels of several FAHFA species, including oleic acid-hydroxy stearic acids (OAHSAs), palmitic acid-hydroxy stearic acids (PAHSAs), and palmitoleic acid-hydroxy stearic

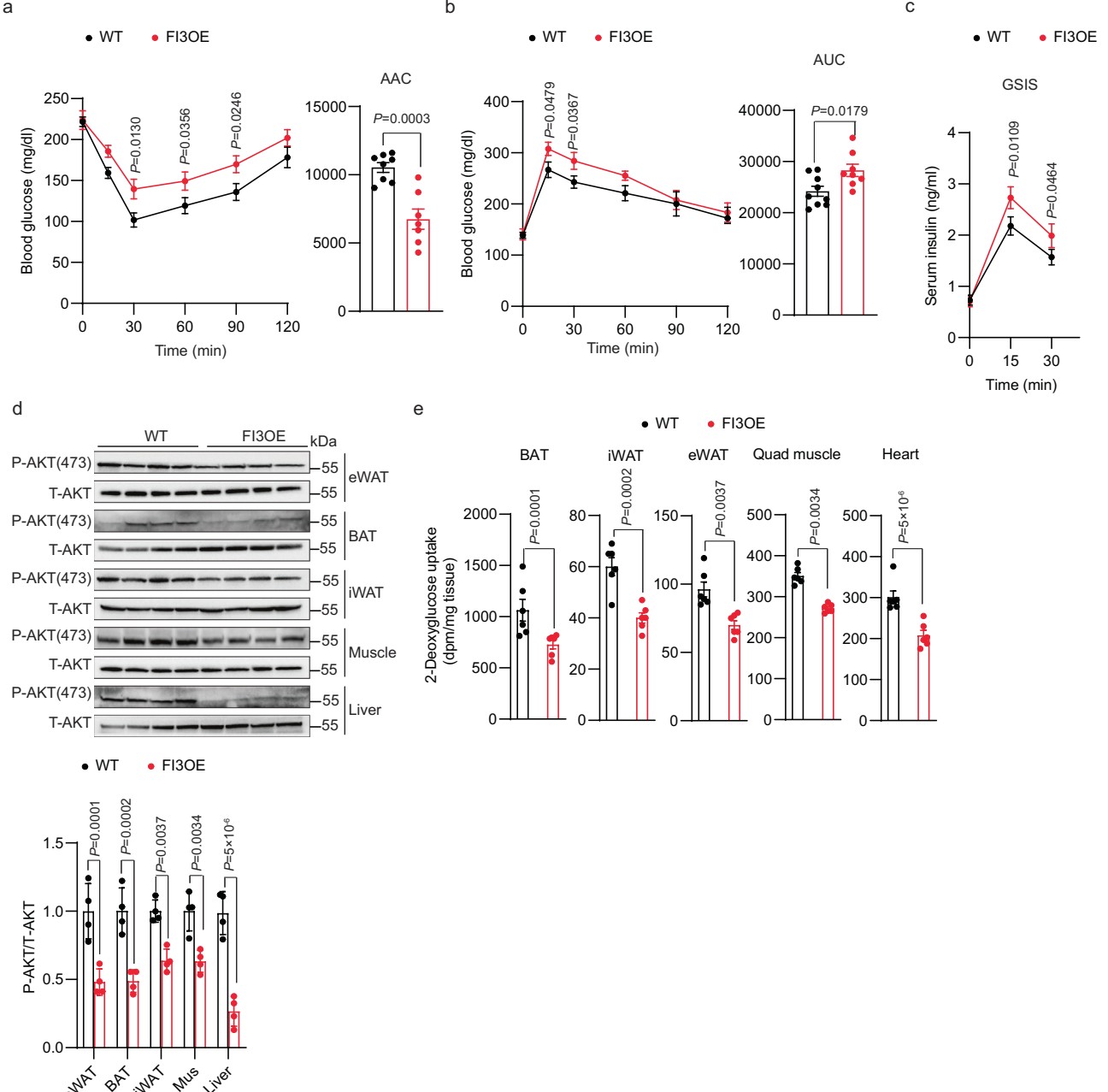

**Fig. 4 | Adipocyte IRF3 promotes HFD-induced insulin resistance at thermoneutrality.** Metabolic analysis of male WT and FI3OE mice ($n = 10$) after 16 weeks of HFD feeding at thermoneutrality, including insulin tolerance test (**a**), glucose tolerance test (**b**), serum insulin levels (**c**), immunoblotting for pAKT (S473) protein levels in eWAT, BAT, iWAT, muscle and liver (**d**), in vivo 2-deoxyglucose uptake in BAT, iWAT, eWAT, quadriceps muscle, and heart (**e**). Statistical significance was assessed by two-way ANOVA (**a**–**c**) or two-tailed Student's *t* test (**d**, **e**). Data in all panels are expressed as mean ± SEM.

acids (POHSAs) (Fig. 6a and Supplementary Fig. 5a). Conversely, reduced FAHFA levels of these same families were observed in FI3OE adipocytes (Fig. 6b and Supplementary Fig. 5b). Underscoring the physiological relevance of these in vitro results, we found that many FAHFA levels were also markedly elevated in vivo in whole eWAT of FI3KO mice (Fig. 6c and Supplementary Fig. 5c), and decreased in the eWAT of FI3OE mice (Fig. 6d and Supplementary Fig. 5d).

## AIG1 is responsible for IRF3-mediated impaired insulin sensitivity

We next tested whether AIG1 is required for IRF3-mediated suppression of insulin sensitivity in vitro. As before, FI3KO adipocytes exhibited elevated p-AKT level and insulin-stimulated glucose uptake, which

was fully suppressible by WT AIG1 overexpression, but not two different alleles of *Aig1* bearing mutations in the catalytic domain (T43A and H134A) (Supplementary Fig. 6a, b)[33]. Conversely, the reduced p-AKT levels and insulin-stimulated glucose uptake observed in FI3OE adipocytes were rescued by knockdown of AIG1 (Supplementary Fig. 6c, d). In addition, decreased glucose uptake was observed in FI3OE SVF-derived adipocytes rescued by FAHFAs (Supplementary Fig. 6e).

To clarify the role of AIG1 in mediating the metabolic functions of IRF3 in vivo, we overexpressed AIG1 via AAV-AIG1 (vs. an AAV-GFP control) in eWAT of WT and FI3KO mice (Fig. 7a). AIG1 overexpression in eWAT did not affect the body weight of WT and FI3KO mice, but largely blunted the metabolic improvements of FI3KO mice (Fig. 7b–e).

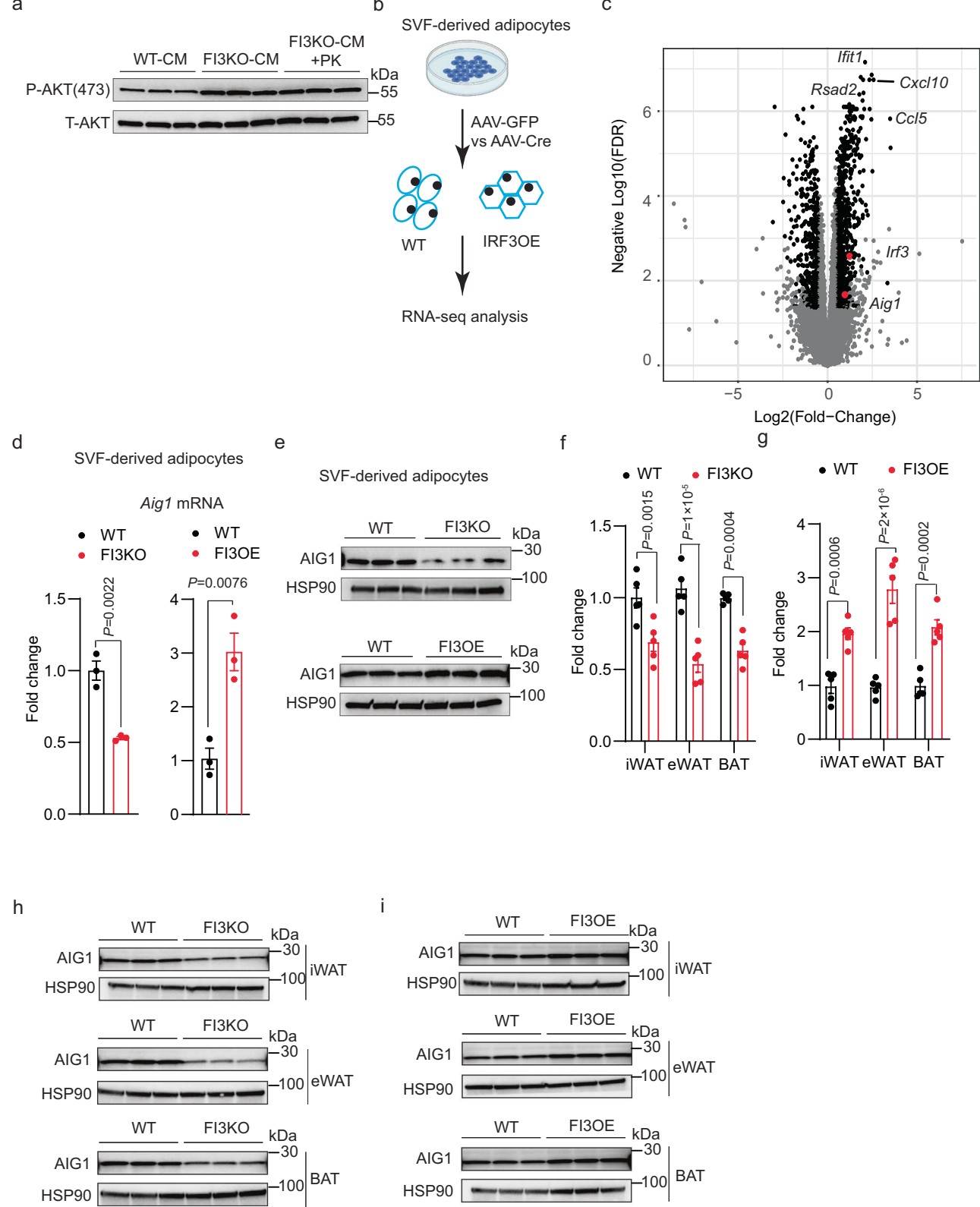

**Fig. 5 | IRF3 increases the transcription of AIG1 in adipocytes. a** Western blot of pAKT (S473) in WT SVF-derived adipocytes pre-treated with indicated conditioned medium plus insulin (100 nM) for 30 min. **b** Scheme showing the experimental paradigm used to identify IRF3 target genes by RNA-sequencing. Created with Biorender.com. **c** Volcano plot of RNA-seq dataset from (**b**). Genes that meet both significance and abundance are shown in black. *Irf3* and *Aig1* are marked in red. mRNA levels of *Aig1* (*n* = 3) (**d**) and western blot (**e**) of AIG1 in WT and FI3KO, FI3OE SVF-derived adipocytes. mRNA levels of *Aig1* in iWAT, eWAT, and BAT from WT, FI3KO (**f**) and FI3OE mice (**g**) (*n* = 5). **h** Western blot of AIG1 in iWAT, eWAT, and BAT of WT and FI3KO mice (*n* = 3). **i** Western blot of AIG1 in iWAT, eWAT, and BAT of WT and FI3OE mice (*n* = 3). Statistical significance was assessed by two-tailed Student's *t* test. Data in all panels are expressed as mean ± SEM.

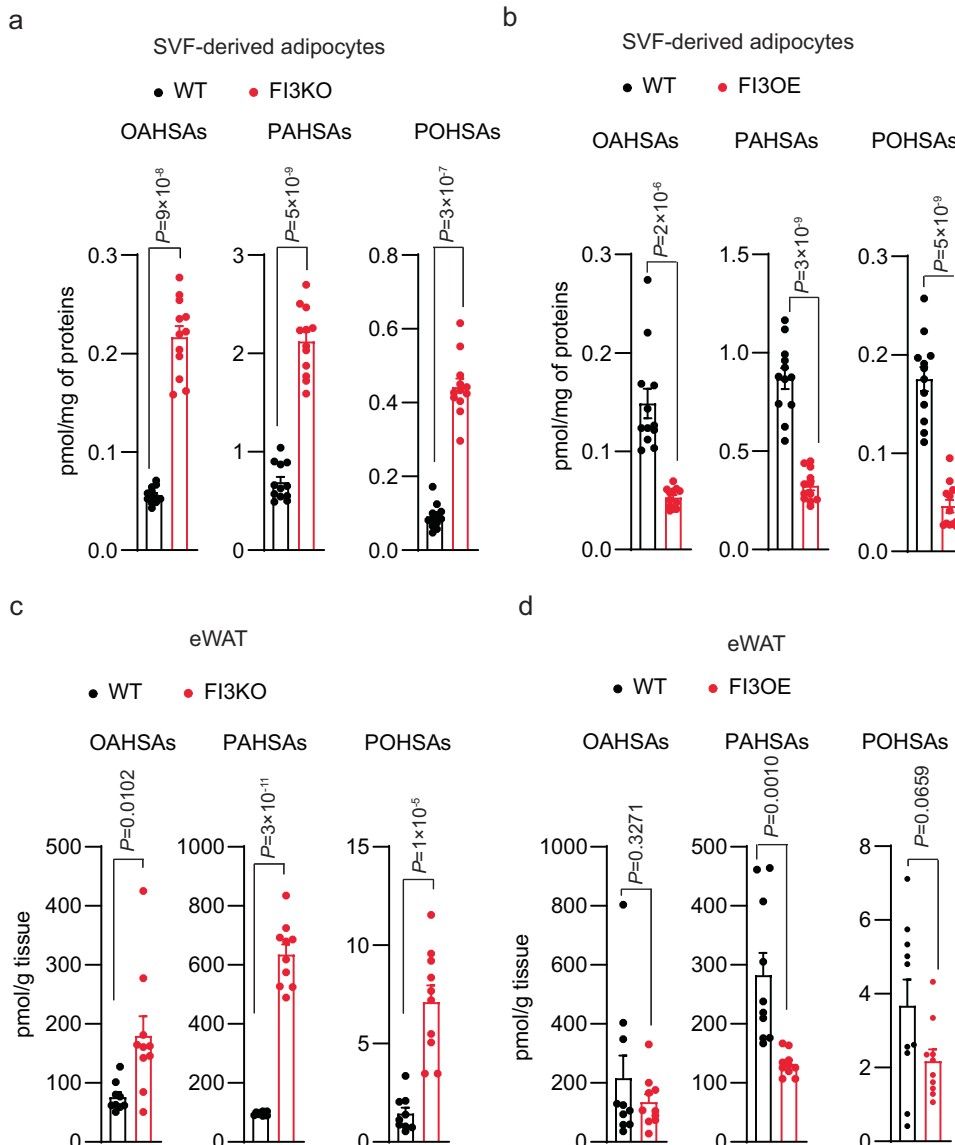

**Fig. 6 | IRF3 decreases intracellular FAHFAs levels. a** Cellular FAHFA levels of WT and FI3KO and **b** WT and FI3OE SVF-derived adipocytes (*n* = 12). **c** Tissue FAHFA levels of eWAT from WT and FI3KO and **d** WT and FI3OE mice (*n* = 10). Statistical significance was assessed by two-tailed Student's *t* test. Data in all panels are expressed as mean ± SEM.

AIG1 overexpression in eWAT also reduced the enhanced p-AKT levels of FI3KO mice in all tissues tested (eWAT, iWAT, BAT, muscle, and liver; Fig. 7f). These data suggest that the adipocyte IRF3-AIG1 axis promotes obesity-induced insulin resistance in mice.

### Pharmacologically inhibiting AIG1 attenuates HFD-induced insulin resistance in FI3OE mice

To explore the therapeutic potential of inhibiting AIG1, we next sought to develop a selective, in vivo-active, AIG1 inhibitor. While a potent, dual AIG1/ADTRP inhibitor (ABD-110207) was previously reported enabling simultaneous in vivo evaluation of these FAHFA hydrolases[34], an in vivo-active AIG1 inhibitor lacking ADTRP activity has yet to be described. Both AIG1 and ADTRP feature an active-site threonine residue which enables both inhibitor screening by activity-based protein profiling (ABPP) using fluorophosphonate (FP) probes and inhibitor design through the application of tailored reactive groups that covalently modify active site nucleophiles[33,34]. Screening of a collection of diverse serine hydrolase-directed compounds by ABPP in mouse brain proteomes, we identified ABD-110000 (called ABD-110 herein) as a

potential AIG1-selective tool compound (Supplementary Fig. 7a). Given the structural similarity of ABD-110 to other covalent serine hydrolase inhibitors[35,36], we believe this compound inactivates AIG1 through carbamylation of the active-site threonine residue (Fig. 8a). Gel-based ABPP profiling of ABD-110 in mouse brain revealed high potency against mAIG1 (IC$_{50}$ = 1.6 nM) and selectivity across other serine hydrolases visible by gel imaging; spiking in ADTRP to these samples further revealed the specificity of the inhibitor (Supplementary Fig. 7b–d). For deeper selectivity profiling across the serine hydrolase family, we employed ABPP-ReDiMe in mouse brain proteomes. Mouse brain lysates were treated first with ABD-110 for 30 min and subsequently with FP-biotin to label active serine hydrolases. Probe-labeled serine hydrolases were then enriched on streptavidin beads and digested with trypsin. The resulting peptides were analyzed by LC-MS/MS following isotopic labeling using reductive dimethylation[34]. ABD-110 potently inhibited AIG1 at both 1 and 10 mM with only a few off-targets detected, including ABHD6 and several highly homologous, mouse-specific carboxylesterases (CESs), which are common targets for carbamates[37]. The high potency of ABD-110 for AIG1 versus these

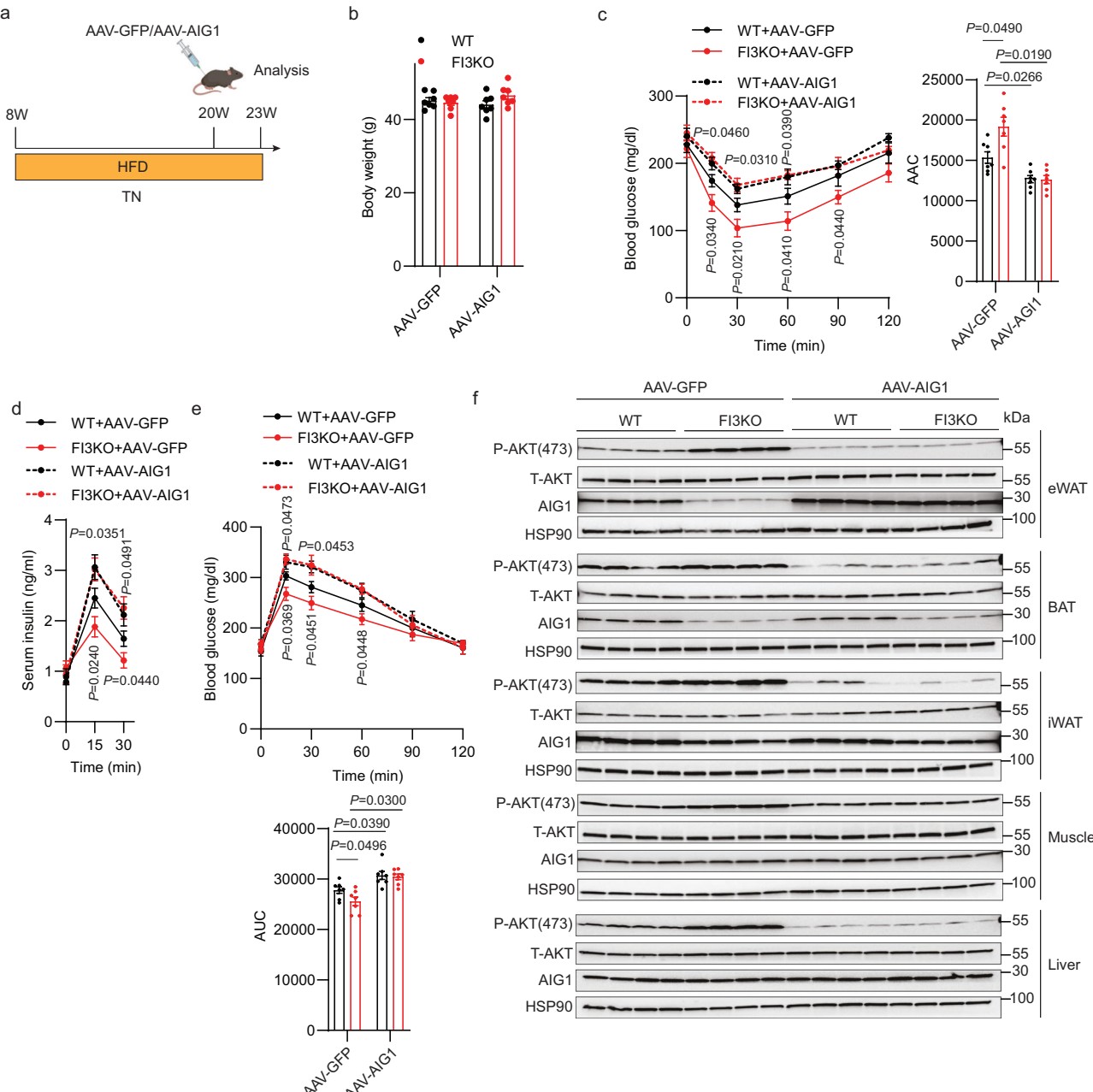

**Fig. 7 | AIG1 is responsible for adipocyte IRF3-mediated suppression of HFD-induced insulin resistance. a** Schematic illustrating the strategy for the AAV injection in eWAT of WT and FI3KO mice on HFD at thermoneutrality. Figure created with Biorender.com. Body weight (**b**), insulin tolerance test (**c**), glucose tolerance test (**d**), and serum insulin levels (**e**) of mice described in (**a**) ($n = 7$–8).

**f** Immunoblotting for pAKT (S473) protein levels in mice's eWAT, BAT, iWAT, muscle, and liver as described in (**a**). Statistical significance was assessed by two-way ANOVA. Data in all panels are expressed as mean ± SEM. *$P < 0.05$ vs WT, #$P < 0.05$ vs. GFP.

off-targets demonstrates the high specificity of ABD-110 for AIG1; similar results were obtained in mouse kidney membranes (Supplementary Fig. 7e-g). AIG1 was not detected in these samples from chow-fed mouse WAT using untargeted MS-based approaches; as shown below, AIG1 was detected in mouse WAT in more directed target engagement studies.

We next evaluated the potential of ABD-110 to be used in vivo. ABD-110 (2.5–25 mg/kg) was administered to C57/BL6J mice by oral gavage and, after 4 h, brains and kidneys were collected to evaluate AIG1 target engagement and selectivity by gel-based ABPP. Brain AIG1 inhibition could be observed at doses as low as 5 mg/kg with maximal inhibition being reached at 10 mg/kg. Notably, we did not observe significant off-target activity for other serine hydrolases or ADTRP with

the exception of CES1c in the kidney which appeared to be maximally inhibited at 5 mg/kg (Supplementary Fig. 7h, i). Finally, we assessed the effect of high-fat feeding on the efficacy of the inhibitor. Chow or high fat-fed mice were dosed with vehicle or ABD-110 (25 mg/kg IP) once daily for 2 weeks. Four hours after the final dose, brain, kidney, eWAT and iWAT tissues were harvested for targeted MS-based ABPP analysis of AIG1 activity. In all tissues tested, the efficacy of AIG1 inhibition was excellent, with slightly improved results in high-fat-fed mice (Supplementary Fig. 7j, k). We speculate that ABD-110 may work better in the setting of obesity because endogenous FAHFAs are reduced[32], providing less competition for binding to AIG1. Together these data support the utility of ABD-110 as a selective AIG1 tool for in vivo applications.

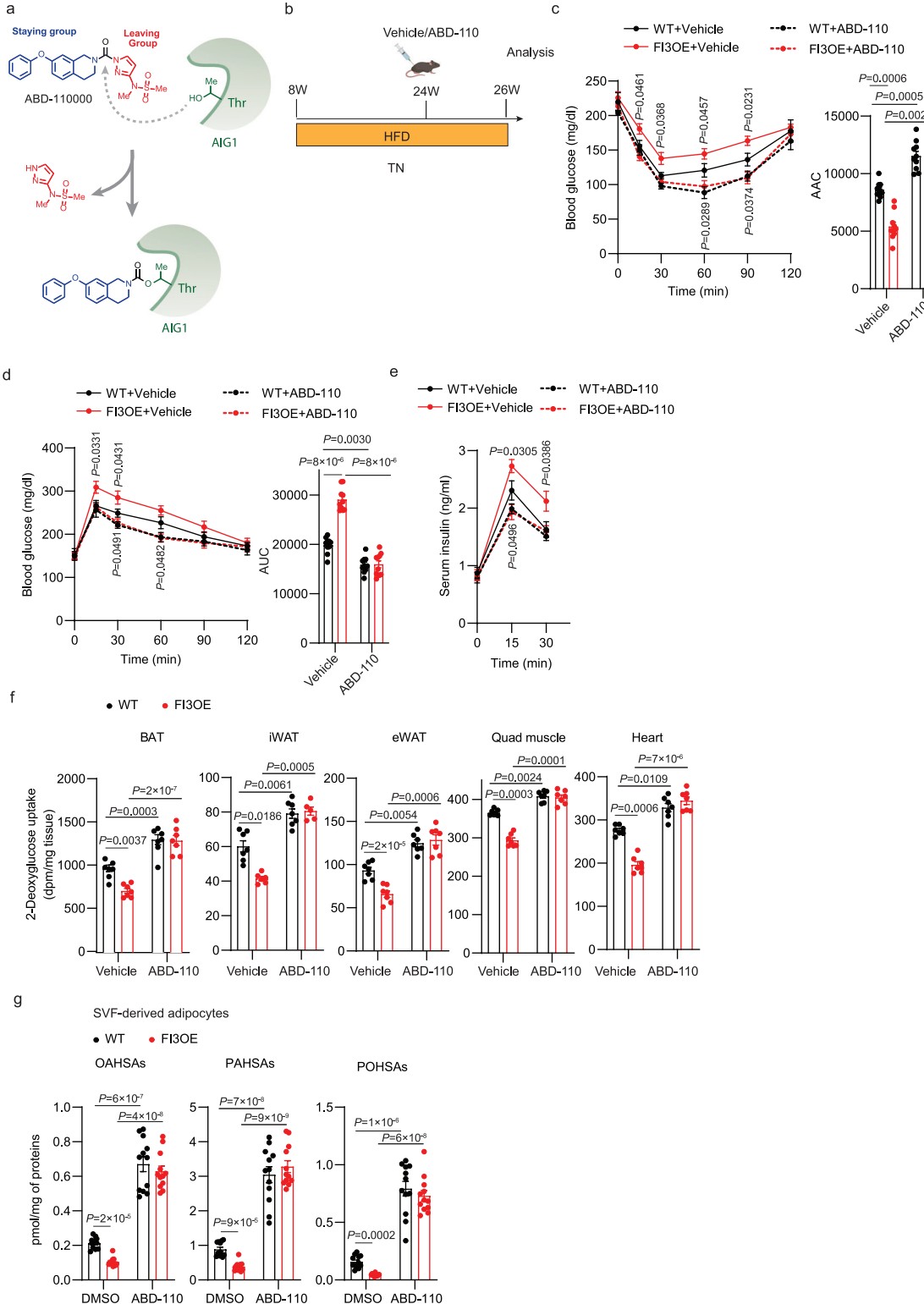

**Fig. 8 | Pharmacologically inhibiting AIG1 attenuates HFD-induced insulin resistance in FI3OE mice. a** Proposed mechanism of covalent AIG1 inhibition by ABD-110. **b** Schematic illustrating the strategy for the AIG1 inhibitor treatment of WT and FI3OE mice on HFD at thermoneutrality. Figure created with Biorender.com. Insulin tolerance test (**c**), glucose tolerance test (**d**), serum insulin levels (**e**), in vivo 2-deoxyglucose uptake in BAT, iWAT, eWAT, quadriceps muscle, and heart (**f**) of mice described in (**b**) (*n* = 7–8). **g** Quantification of FAHFA levels in WT and FI3OE SVF-derived adipocytes treated with or without ABD-110 (1 μM, 4 h) (*n* = 12). Statistical significance was assessed by two-way ANOVA. Data in all panels are expressed as mean ± SEM. *$P$ < 0.05 vs WT, #$P$ < 0.05 vs vehicle.

To assess the effect of ABD-110 on metabolic function, chow and high-fat-fed WT mice were injected with vehicle or 25 mg/kg ABD-110 per day for 2 weeks. ABD-110 had no effect on insulin sensitivity and glucose tolerance on chow diet (Supplementary Fig. 8a–c). However, the impaired insulin and glucose tolerance of high fat-fed mice were ameliorated by ABD-110 treatment (Supplementary Fig. 8d–f). We next asked whether ABD-110 would rescue the metabolic dysfunction of FI3OE mice. High fat-fed WT and FI3OE mice were injected with vehicle or 25 mg/kg ABD-110 per day for 2 weeks, with no difference in body weight noted after ABD-110 injection (Fig. 8b, Supplementary Fig. 9a, b). The impaired insulin and glucose tolerance of high fat-fed FI3OE mice were ameliorated by AIG1 inhibition (Fig. 8c, d). Furthermore, the increased glucose-stimulated insulin secretion observed in FI3OE mice after glucose injection was reversed by ABD-110 treatment (Fig. 8e). Consistent with this, treatment with ABD-110 increased insulin-stimulated glucose uptake in all tissues tested in both WT and FI3OE mice and abolished the effects of adipocyte IRF3 overexpression (Fig. 8f). As before, FI3OE adipocytes exhibited reduced FAHFA levels. ABD-110 increased FAHFA levels in WT adipocytes and fully reversed the decrease in FI3OE adipocytes (Fig. 8g, Supplementary Fig. 9c).

## Discussion

Overnutrition and obesity promote a pro-inflammatory milieu in adipose tissue and other metabolically relevant tissues, and key extracellular mediators of inflammation, including cytokines like TNFα and IL-1β, and TLR ligands like LPS and cell-free DNA, are well established as endogenous drivers of insulin resistance[38–41]. Despite years of intensive study, there is still an incomplete understanding of how inflammatory signaling causes insulin resistance. Several proposed mechanisms involve direct cross-talk between inflammatory kinases like JNK and IKK and various insulin signaling components[42,43]. However, the time course over which insulin action is affected by inflammatory factors is much longer than would be expected if this were a primary mechanism[8,24]. This suggests a potential role for the transcriptional effects of inflammation, a notion supported by the fact that alterations in chromatin state are known to affect insulin sensitivity, and also the proven utility of transcription factor-based insulin sensitizers[44,45]. To date, however, few transcriptional pathways have been convincingly connected to insulin resistance. Among those that have been described, expression of *Slc2a4* (encoding the insulin-sensitive glucose transporter GLUT4) is reduced in adipocytes from insulin resistant humans, and targeted ablation of GLUT4 in adipocytes of mice resulted in insulin resistance in adipocytes, liver and muscle with reduced total body insulin sensitivity measured by euglycemic clamp studies[46]. In addition, expression of *Mlxipl* (encoding Carbohydrate Response Element Binding Protein, ChREBP) is reduced in adipose tissue of insulin resistant people[47] and adipose specific knockout of ChREBP also results in insulin resistance in adipocytes, liver and muscle[48]. Finally, reduced *Pparg* expression has been noted in adipocytes from insulin resistant humans and rodents[46,49], but whether this causes insulin resistance is not known. We have provided evidence that the pro-inflammatory transcription factor IRF3 is a critical driver of insulin resistance in the context of obesity. IRF3 levels are increased in the adipocytes of people and mice with obesity, and mice lacking IRF3 in all tissues show improved insulin sensitivity before body weight divergence after high fat feeding[24]. We have also shown that IRF3 becomes activated in the hepatocytes and adipocytes of people and mice with obesity, where it mediates cellular and systemic insulin resistance[25,26]. Interestingly, in hepatocytes, IRF3 drives insulin resistance by directly inducing the expression of *Ppp2r1b*, a component of the PP2A phosphatase complex[26]. The IRF3-mediated increase in PP2A activity diminishes AMPK and AKT phosphorylation, with subsequent impairment of glucose homeostasis. Here, we find that IRF3 is an inducer of *Aig1* in adipocytes, which reduces levels of insulin-

sensitizing FAHFAs. Although the specific gene targets of IRF3 differ between these two cell types, both illustrate the key finding that pro-inflammatory factors ultimately affect insulin action through induction of genes that are outside the traditional inflammatory gene portfolio. Importantly, our data do not rule out potential roles for other downstream targets of IRF3, or for pro-inflammatory transcription factors besides IRF3 (AP-1 factors, for example). The novelty of the current work lies in (a) demonstrating that the adipocyte is a key cell type mediating the effect of IRF3 on insulin action, (b) showing definitively that the effect on insulin sensitivity is independent of weight changes, and most importantly, (c) identifying a specific molecular link between IRF3 and insulin resistance (i.e., reduction of FAHFAs by promoting expression of a FAHFA hydrolase).

Some FAHFAs, in particular 5- and 9-PAHSA, have been shown to exert a myriad of beneficial metabolic actions, including improvement in insulin sensitivity and in glucose-stimulated insulin secretion. Many FAHFAs also exert anti-inflammatory effects in vitro, and 5- and 9-PAHSA protect islets from the effects of autoimmunity[50] in vivo and also delay the onset and reduce the severity of experimental colitis by modulating both innate and adaptive immune responses[51]. Our work illustrates that the converse is also true; inflammation leads to reduced FAHFA levels, at least in the context of overnutrition. Recently, ATGL has been shown to catalyze the formation of the FAHFA ester bond in mammals through its transacylase activity[52]. FAHFA degradation, on the other hand, is mediated by at least three hydrolases, AIG1 and ADTRP[34], and carboxyl ester lipase[53], the last of which is not expressed in adipocytes[54]. Insulin resistant mice and humans have reduced FAHFA levels in serum and subcutaneous adipose tissues[32], although it has been unclear how this occurs. Here, we have shown that *Aig1* is a transcriptional target of IRF3 in adipocytes, and targeted lipidomic analysis from adipocytes identifies reductions in FAHFA levels as a consequence of IRF3-mediated inflammation. IRF3-mediated suppression of systemic glucose uptake was rescued by AIG1 inhibition, which is consistent with the previous finding that 5-PAHSA primes adipocytes for de novo lipogenesis[55]. and multiple FAHFA isomers directly augment insulin-stimulated glucose uptake in adipocytes[32,56] by augmenting GLUT4 translocation to the plasma membrane[32]. Interestingly, targeted ablation of AIG1 in all tissues failed to protect mice from HFD-induced insulin resistance and glucose intolerance[34]. The rationale for these conflicting results is unclear, and should be explored further.

A limitation of our study is that we have not determined whether the changes in systemic insulin sensitivity seen in FI3KO and FI3OE animals result from altered serum FAHFA levels, or whether the effect may be autocrine or paracrine. Our initial data indicate that serum POHSA levels, but not PAHSAs or OAHSAs, are elevated in serum of FI3OE mice while all three families are elevated in serum of FI3KO mice (Supplementary Fig. 10). Multiple factors regulate serum FAHFA levels and it is beyond the scope of this paper to determine the mechanisms underlying these increased levels. However, it is important to note that adipose tissue and serum FAHFA levels can be discordant. For example, fasting increases adipose FAHFA levels, while simultaneously reducing levels in the serum[32]. Importantly, local insulin sensitization from increased FAHFA concentrations in adipose tissue itself can reduce systemic insulin resistance, partly mediated by reduced serum insulin. Moreover, the potent anti-inflammatory actions of FAHFAs may reduce intra-adipose cytokine production, with beneficial consequences for distal metabolism. Furthermore, POHSAs, PAHSAs, and OAHSAs have been shown to have important biological effects in cell-based assays[56]. While in vivo administration of 5- and 9-PAHSA enhances insulin sensitivity at the whole-body level[57], 12-OAHSA had a more modest effect[56] and no data are available for effects of in vivo administration of POHSAs. The physiological role of local vs. circulatory FAHFA levels requires further inquiry.

## Methods

### Animal studies

All animal experiments at BIDMC were performed with approval from the Institutional Animal Care and Use Committees of the Harvard Center for Comparative Medicine and Beth Israel Deaconess Medical Center in an AAALAC-accredited animal facility in accordance with the guidelines outlined by the Guide for the Care and Use of Laboratory Animals and the Animal Welfare Act. Mice were maintained on a chow diet (Harlan Teklad, 8664) under a 12 h light/dark cycle at 22 °C for standard room temperature housing. Eight-week-old male mice were fed a 60% high-fat diet (D12492, Research Diets) at thermoneutrality for 16 weeks and body weight was measured weekly. All metabolic studies were performed in male mice. Strains utilized include C57BL/6 (JAX:000664), IRF3 floxed mice (JAX:036260), IRF3-2D mice (JAX:036261) and *Adipoq-Cre* (JAX:028020). Mice at BIDMC were euthanized using $CO_2$ inhalation. All animal experiments at Lundbeck La Jolla Research Center were performed at an AAALAC-accredited animal facility in accordance with the guidelines outlined by the Guide for the Care and Use of Laboratory Animals and the Animal Welfare Act and were approved by the Explora Biolabs Institutional Animal Care and Use Committee (IACUC). Animals were maintained under a 12-h/12-h light/dark cycle and allowed free access to food and water when residing in home cages and for the duration of the study. LLJRC study performed used female C57Bl/6 J mice of 10 weeks of age from Jackson Labs. At the indicated time during the study, mice were anesthetized with isoflurane and decapitated prior to collection of relevant tissues for analysis of target engagement.

### Local delivery of adeno-associated virus (AAV) in adipose tissues

Adeno-associated virus serotype 8 expressing mouse Aig1 and AAV8-GFP control were purchased from Applied Biological Materials (Canada). AAV injection was carried out in mice anesthetized with an intraperitoneal injection of ketamine (100 mg/kg) and xylazine (10 mg/kg)[58]. A laparotomy was performed to expose eWAT. Each eWAT pad received four injections of 10 μL AAV solution using a Hamilton syringe to distribute the vector over the whole depot. The abdomen was rinsed with sterile saline solution and closed with a two-layer suture. Mice were monitored for changes in body weight and subjected to further analysis.

### Glucose tolerance tests (GTT) and insulin tolerance tests (ITT)

For glucose tolerance tests (GTT), mice were fasted overnight and then injected intraperitoneally with glucose (1.5 g per kg body weight). For insulin tolerance tests (ITT), mice were fasted for 6 h and then injected intraperitoneally with insulin (Humulin, Eli Lilly; 1.5 U per kg body weight). Blood samples were collected from a tail nick at the indicated time points, and glucose levels were measured using test strips (OneTouch, USA).

### Serum insulin quantification

Blood samples from fed mice were centrifuged at 3000 *g* for 10 min, and plasma was pipetted and stored at 80 °C. Serum insulin levels were measured using the ultra-sensitive mouse insulin ELISA kit according to the manufacturer's instructions (Crystal Chem, USA).

### Mouse and human pre-adipocyte culture and differentiation

Inguinal subcutaneous WAT stromal-vascular fraction (SVF) cells were isolated from both male and female mice, cultured, and differentiated into mature adipocytes[59]. Inguinal white fat pads were obtained from male mice (4–6 weeks), and then dissected, washed three times with phosphate-buffered saline (PBS), minced into small pieces, and then digested for 45 min in digestion buffer [1 mg/mL collagenase I (Sigma) and 1% bovine serum albumin (BSA) (Sigma-Aldrich) and 10 mM $CaCl_2$ (ThermoFisher) in PBS] at 37 °C. Digested tissues were passed through a sterile 100 μm cell strainer (BD Falcon) to remove undigested

fragments and centrifuged at 600 *g* for 5 min at 4 °C to pellet the SVF cells. The pellet was resuspended in DMEM/F12+ GlutaMax (Invitrogen) medium containing 1% pen/strep and 10% FBS, then passed through a 40 μm cell strainer, centrifuged, resuspended as described above, and plated on collagen-coated culture dishes (Corning). For differentiation, SVF cells were grown to confluence and induced (day 0) in differentiation medium supplemented with 5 μg/ml insulin (Sigma-Aldrich, I9278), 0.5 mM 3-isobutyl-1-methylxanthine (IBMX, Sigma-Aldrich, I5879), 5 μM dexamethasone (Sigma-Aldrich, D4902), and 1 μM rosiglitazone (Sigma-Aldrich, R2408) for 2 days and subsequently cultured in maintenance medium supplemented with 5 μg/ml insulin and 1 μM rosiglitazone. On day 6, the SVF-derived adipocytes were transduced with 2 μl AAV-GFP or AAV-GFP-Cre for 12 h. Experiments were carried out on day 8 of differentiation.

Human SGBS preadipocytes were seeded in a 12-well plate at $8 \times 10^4$ per well and cells were cultured in a normal culture medium (DMEM/F12+ GlutaMax with 1% pen/strep and 10% FBS) until they reached confluency. Cells were incubated in normal culture medium (DMEM/F12+ GlutaMax with 1% pen/strep and 10% FBS) with 4 ng/ml human FGF (Perprotech, 100-18B) for 2 days, then incubated (day 0) in differentiation medium [1% pen/strep, 33 nM Biotin (Sigma-Aldrich, B-4639), 17 nM Panthotenate (Sigma-Aldrich, P-5155), 10 μg/ml Transferrin (Sigma-Aldrich, T-2252), 20 nM insulin, 100 nM cortisol (Sigma-Aldrich, H-0888), 0.2 nM T3 (Sigma-Aldrich, T-6397), 25 nM dexamethasone, 250 μM IBMX and 2 μM rosiglitazone in DMEM/F12+ GlutaMax] for 4 days. After the fourth day, the cells were incubated in the maintenance medium (1% pen/strep, 33 nM, 17 nM Panthotenate, 10 μg/ml Transferrin, 20 nM insulin, 100 nM cortisol, and 0.2 nM T3 in DMEM/F12+ GlutaMax); maintenance medium was changed every fourth day. Experiments were carried out on days 10-day and 12.

### RNA-Seq

For RNA-seq analysis, sequencing reads were demultiplexed and trimmed for adapters using bcl2fastq (v2.20.0). Secondary adapter trimming, NextSeq/Poly(G) tail trimming, and read filtering was performed using fastp (v0.20.1)[60]; low-quality reads and reads shorter than 24 nt after trimming were removed from the read pool. Salmon (v1.4.0)[61] was used to simultaneously map and quantify reads to transcripts in the GENCODE M24 genome annotation of GRCm38/mm10 mouse assembly. Salmon was run using full selective alignment, with sequence-specific and fragment GC-bias correction turned on (--seqBias and --gcBias options, respectively). Transcript abundances were collated and summarized to gene abundances using the tximport package for R[62]. Normalization and differential expression analysis were performed using edgeR. For differential gene expression analysis, genes were considered significant if they passed a fold change (FC) cutoff of log2FC ≥ 1 and a false discovery rate (FDR) cutoff of FDR ≤ 0.05. This paper's accession number for the RNA-seq datasets is GEO: GSE213048.

### RNA isolation/quantitative RT-PCR

Total RNA was extracted from cells or tissues using a Direct-zol RNA MiniPrep kit (ZYMO Research, USA) or Trizol reagent (Invitrogen, USA). Reverse transcription was performed with 1 μg of total RNA using a High-Capacity cDNA Reverse Transcription Kit (Thermo Fisher, USA). qRT-PCR was performed on an ABI PRISM 7500 (Applied Biosystems, USA). Melting curve analysis was carried out to confirm the RT-PCR products. Statistical analysis was performed using the ΔΔCT method with TBP primers as control (primer sequences in Supplementary Table 1).

### Immunoblotting

For immunoblotting analyses, tissues and cells were lysed in RIPA buffer containing protease and phosphatase inhibitors (Thermo Fisher, USA). Protein levels were quantified using a BCA protein assay

kit (Thermo Fisher, USA) and lysates containing an equal amount of protein were subjected to SDS-PAGE and transferred to polyvinylidene fluoride (PVDF) membranes, followed by incubations with primary and secondary antibodies. Primary antibodies against phospho AKT (Ser473) (CST, 9271), AKT (AKT, 4685), HSP90 (CST, 4877), phospho IRF3(Ser396) (CST, 29047), IRF3 (CST, 11904) and AIG1 (Proteintech, 14468-1-AP), were used.

## Glucose uptake assays

Glucose uptake assays were performed as described earlier[24]. Briefly, SVF-derived adipocytes (day 6) and human SGBS cells (day 14) were incubated in serum-free DMEM/F12 for 4–6 h. Cells were then washed three times with KRH buffer (12 mM Hepes, pH 7.4, 121 mM NaCl, 5 mM KCl, 0.33 mM CaCl$_2$, and 1.2 mM MgSO$_4$) and incubated for 20 min in KRH buffer in the absence or presence of 100 nM insulin. Cells were treated with 2-deoxy-d-[2,6-$^3$H]-glucose (0.33 µCi/ml) for another 10 min. Glucose uptake was stopped quickly by three rapid washes with KRH buffer containing 200 mM glucose and 10 µM cytochalasin B on ice. Cells were solubilized in 0.1% SDS for 30 min, and radioactivity was measured by liquid scintillation counting. In some experiments, a combination of 9-POHSA (20 µM), 9-PAHSA (20 µM) and 9-OAHSA (20 µM) (Cayman Chemical), or Vehicle (DMSO, 0.01%) was incubated with SVF-derived adipocytes for 24 h, then insulin-stimulated glucose uptake studies were initiated. Total protein was determined by the BCA method (Pierce), and results were normalized to protein amount.

## Insulin signaling in vivo

Mice were fasted overnight for insulin signaling studies, and insulin (10 U/kg body weight) was administered i.p. After 10 min various tissues were harvested and stored at −80 °C until use. Tissue samples were homogenized in cell signaling lysis buffer containing protease inhibitors (Roche) and phosphatase inhibitors (Sigma-Aldrich) and subjected to Western blotting.

## Construct design and site-directed mutagenesis

To construct the mouse AIG1 overexpression lentivirus, the open reading frames of mouse *Aig1* was cloned into pCDH-CMV-MCS-EF1-Puro cDNA Cloning and Expression Vector. To construct the shAIG1 lentivirus, the short hairpins (5′′-ACCTTCTCCGTGGGCTATATA-3′ and 5′- CTATGACAGAGAGATGATATA-3′) were cloned into the pLKO.1-TRC cloning vector, respectively. To construct shIRF3 lentivirus, the short hairpin (5′-CGAAGTTATTTGATGGCCTGA-3′) was cloned into the pLKO.1-TRC cloning vector. To generate dox-inducible IRF3 overexpression lentivirus, mouse IRF3 cDNA was cloned into pCW57-MCS1-P2A-MCS2 (Hygro). To construct dox-inducible shIRF3 lentivirus, the short hairpin (5′-CGAAGTTATTTGATGGCCTGA-3′) was cloned into the Tet-pLKO-puro cloning vector. Mutagenesis primers used to construct AIG1(T43A) were as follows: forward, 5′- GAAGTTCCTGGCCTTCATT-GATC-3′; reverse, 5′- GATCAATGAAGGCCAGGAACTTC-3′; mutagenesis primers for AIG1(H134A): forward, 5′- CCATGGAATGG CCACAACGGTTT-3′; reverse, 5′- AAACCGTTGTGGCCATTCCATGG-3′.

## Lentivirus packaging

psPAX2 and pMD2.G were used for the lentiviral packaging. High-titer lentivirus was packaged in 293T cells using lipofectamine 2000 transfection agent. Viral supernatants were collected 48 h post-transfection and used for pre-adipocyte infection.

## Tissue glucose uptake in vivo

In vivo tissue glucose uptake studies were carried out with mice maintained on HFD in thermoneutrality. After a 4-h fast, mice were injected with insulin (Humulin, 0.75 U/kg, ip) and a trace amount of 2-deoxy-D-[1-$^{14}$C] glucose (10 µCi; PerkinElmer). Forty minutes later, the mice were euthanized, and tissues were collected, weighed, and homogenized. Radioactivity was measured and counted as described[63].

## Lipid extraction for FAHFA analysis

For the measurement of FAHFAs from SVF-derived cells, total lipids were extracted using the modified Bligh–Dyer method[64]. In brief, 1.5 ml of 2:1 chloroform/methanol with the internal standards [$^{13}$C$_{16}$] 9-PAHSA, [$^{13}$C$_{18}$] 12-OAHSA, and [$^{13}$C$_{16}$] 5-PAHSA was added to 500 µl of cell suspension in PBS. Samples were vortexed and centrifuged at 2000 *g* for 7 min. The bottom organic phase was transferred into a new vial and dried under an N$_2$ stream.

To extract total lipid from white adipose tissue, eWAT (~50 mg) was dounce homogenized on ice in a mixture of 1.5 ml PBS, 1.5 ml methanol and 3 ml chloroform. Internal standards [$^{13}$C$_{16}$] 9-PAHSA, [$^{13}$C$_{18}$] 12-OAHSA, and [$^{13}$C$_{16}$] 5-PAHSA were added to chloroform prior to extraction. The resulting mixture was centrifuged at 2000 *g* for 7 min and the bottom organic phase was transferred into a new vial and dried under a N$_2$ stream[52]. To extract total lipids from mouse serum samples, PBS was added to serum (200 µl) such that the final volume was 1.5 mL. Subsequently, methanol and chloro-form were added to this mixture (PBS/methanol/chloroform ratio was 1:1:2). The resulting mixture was shaken for 30 s by hand, vortexed for 15 s, and then centrifuged to separate the organic and aqueous phases. The organic phase containing lipids was then subjected to solid-phase extraction[57].

For eWAT and serum samples, FAHFAs were enriched using solid-phase extraction (SPE) columns (Hypersep silica 500 mg)[65]. Columns were equilibrated with 15 ml hexane, and the samples were then resuspended in 200 µl chloroform and loaded on the SPE column. The neutral lipid fraction containing TGs was eluted with 16 ml of 95:5% hexane/ethyl acetate, and the polar lipid fraction containing non-esterified FAHFAs was eluted using 15 ml ethyl acetate. The FAHFA fractions were dried under a stream of N$_2$ and stored at −40 °C until LC-MS/MS analysis.

## FAHFA analysis using LC−MS/MS

FAHFA isomers were quantified using an Agilent 6470 Triple Quad LC−MS/MS instrument through multiple reaction monitoring (MRM) in the negative ionization mode[52]. In brief, cell culture lipid extracts were reconstituted in 50 µl methanol, and tissue extracts were reconstituted in 100 µl methanol. 7 µl of each sample were injected into a UPLC BEH C18 Column (Waters Acquity, 186002352). FAHFA regioisomers were resolved using 93:7 methanol/water with 5 mM ammonium acetate and 0.01% ammonium hydroxide solvent through an isocratic gradient at 0.15 ml min$^{-1}$ flow rate for 60 min. Transitions for endogenous PAHSAs were m/z 537.5 → m/z 255.2 (CE = 30 V) and m/z 537.5 → m/z 281.2 (CE = 25 V), and the transition for $^{13}$C$_{16}$-9-/$^{13}$C$_{16}$-5-PAHSAs was m/z 553.5 → m/z 271.3 (CE = 30 V). Transitions for endogenous OAHSAs were m/z 563.5 → m/z 281.2 (CE = 30 V) and m/z 563.5 → m/z 299.3 (CE = 23 V), and the transition for $^{13}$C$_{18}$-12-OAHSA was m/z 581.6 → m/z 299.3 (CE = 30 V). Transitions for endogenous POHSAs were m/z 535.5 → m/z 299.3 (CE = 30 V), m/z 535.5 → m/z 253.2 (CE = 25 V), m/z 535.5 → m/z 271.3 (CE = 30 V), and m/z 535.5 → m/z 281.2 (CE = 25 V). MS acquisition parameters for tandem MS were: gas temperature = 250 °C, gas flow = 12 l min$^{-1}$, nebulizer = 20 psi, sheath gas temperature = 250 °C, sheath gas flow = 11 l min$^{-1}$. Spray voltage was −1.0 kV.

Each FAHFA regioisomer levels were quantified by normalizing their peak area (extracted using MassHunter 10.0) to the internal standard [$^{13}$C$_{16}$]9-PAHSA peak area and total protein amount or tissue weight and multiplied for the amount of internal standard added. OAHSAs were quantified using the $^{13}$C$_{18}$-12-OAHSA as internal standard.

Lipidomic samples and workflow are described further in Supplemental Note 1.

## ABD-110 in vivo dose response

Female C57Bl/6J mice (Jackson) aged 10 weeks at the time of dosing were administered ABD-110 by oral gavage (10 ml/kg volume). ABD-110 was prepared fresh on the day of dosing in water containing 5% solutol and 20% HpbCD. Maximal dispersal of the compound was achieved by bath and probe sonication until a uniform white suspension was formed. Animals were administered single oral doses of ABD-110 (2.5–25 mg/kg). Four hours after compound administration, animals were anesthetized with isoflurane and decapitated. Brains and kidneys were removed and rinsed in PBS before freezing in liquid nitrogen. Tissues were stored at −80 °C until analysis by gel-based ABPP (see below).

## Proteome preparation for ABPP

**Mouse tissue (brain, kidney, and adipose) homogenates.** Frozen mouse tissues (one brain hemisphere, one whole kidney or -100 mg of WAT) were homogenized by adding the tissue into a 2 mL Safe-lock microcentrifuge tube along with 0.5–1.0 mL of PBS and a 5 mm steel bead. The tissue was then homogenized using the Qiagen TissueLyzer at 30 Hz for 1 min. After homogenization, samples were put through a clarifying spin (1000 $g$, 10 min, 4 °C) and supernatants were collected.

Brain and kidney homogenates were fractionated further by ultracentrifugation (100,000 $g$, 45 min, 4 °C) and membrane pellets were resuspended in PBS. Protein concentrations were determined using Bio-Rad DC protein assay and diluted to desired protein concentration (see below) for subsequent ABPP analysis.

## Determination of ABD-110 potency and selectivity by ABPP

**Gel-based ABPP.** Inhibitor potency ($IC_{50}$ values and target engagement) against AIG1 and other serine/threonine hydrolases was determined by competitive gel-based ABPP in mouse brain and kidney membrane homogenates using FP-Rh competition[66]. Mouse brain and kidney proteomes were prepared as described above. For determination of mADTRP activity, full-length mADTRP (Dharmacon) was recombinantly expressed in HEK293T cells and the corresponding cell lysates were doped into mouse brain membrane proteomes (1.0 mg/mL) at a ratio to enable detection of both AIG1 and ADTRP.

In vitro potency for AIG1, ADTRP and other serine hydrolase enzymes was determined by treating brain proteomes (50 µg, 1.0 mg/mL with or without mADTRP-containing cell lysates) with ABD-110 or DMSO for 30 min at 37 °C and subsequently treated with FP-Rh (1.0 µM) for an additional 30 min at room temperature.

For gel-based in vivo target engagement, brain and kidney proteomes (50 µg) from vehicle or ABD-110-treated mice were treated with FP-Rh (1.0 µM) for 30 min at room temperature.

After incubation with FP-Rh, reactions were quenched with 4X SDS-PAGE loading buffer and FP-Rh-labeled enzymes were resolved by SDS-PAGE (10% acrylamide). In-gel fluorescence was visualized using a Bio-Rad ChemiDoc™ XRS imager. Fluorescence is shown in gray scale. Quantification of enzyme activities was performed by densitometric analysis using ImageJ software (NIH). Integrated peak intensities were generated for the band corresponding to AIG1[34]. $IC_{50}$ values were calculated through curve-fitting semi-log-transformed data ($x$-axis) by non-linear regression with a four-parameter, sigmoidal dose response function (variable slope) in Prism software (GraphPad).

## MS-based ABPP

Brain, kidney and adipose tissue proteomes (2.5 mg/mL in 0.2 mL of PBS) were treated with inhibitor at the specified concentrations (0.001–10 µM) or DMSO for 30 min at 37 °C and subsequently labeled with FP-biotin (10 µM) for 1 h at room temperature. Proteome samples were immediately precipitated and processed through the MS-based ABPP sample preparation protocol (below).

## MS-based ABPP sample preparation

Tissue proteomes (2.5–5 mg/mL normalized by tissue source) in 200 mL of PBS were labeled with FP-biotin (50 µM) for 1 h at 37 °C. After labeling, the proteomes were denatured and precipitated using 9:1 acetone/MeOH, resuspended in 0.2 mL of 8 M urea in PBS and 1% SDS, reduced using DL-dithiothreitol (DTT, 10 mM) for 20 min at 55 °C, and then alkylated using iodoacetamide (50 mM) for 30 min at room temperature in the dark. The biotinylated proteins were enriched with PBS-washed streptavidin-agarose beads (50 µL; Thermo Scientific) by rotating at room temperature for 1.5 h in PBS with 0.2% SDS (1.3 mL). The beads were then washed sequentially with 0.5 mL 0.2% SDS in PBS (10×), 1.1 mL PBS (10×) and 1.1 mL DI $H_2O$ (10×). On-bead digestion was performed using sequencing-grade trypsin (2 µg; Promega) in 2 M urea in PBS for 12–14 h at room temperature (100 µL). The beads were removed using filtration and washed with DPBS (100 µL).

For in vitro MS-based ABPP selectivity profiles in mouse brain and kidney, peptide digests derived from vehicle (DMSO) and ABD-110-treated proteomes were dimethylated with the addition of light ($CH_2O$) or heavy ($^{13}CD_2O$) formaldehyde (0.15%) and $NaBH_3CN$ (22.2 mM)[67]. After incubating for 1 hr at room temperature, the methylation reaction was quenched with $NH_4OH$ (40 µL, 1%) and, subsequently, formic acid (20 µL, 100%) before the heavy and light samples were combined. The mixed samples were then desalted as described below. For all other samples, the digests were acidified by addition of formic acid (10 µL of 100% formic acid) and desalted using SOLAµ™ SPE Plates (HRP 2 mg/1 mL). Samples were dried by centrifugal evaporation. In vitro ABD-110-treated WAT samples were labeled with TMTpro reagents (see below) whereas in vivo ABD-110-treated samples were stored at -80 °C until targeted MS analysis.

## TMTpro labeling protocol

TMTpro labeling protocols were adapted from previous work[68]. Dried samples were reconstituted in 20 µL EPPS buffer, pH 8.5 followed by addition of 5 µL 40 mM TMTpro reagents in acetonitrile for 1 hr. Reactions were quenched with 5 µL 5% hydroxylamine, mixed, desalted, dried as in prior section, and stored at −80 °C until analysis.

## MS data acquisition and analysis

**Parallel reaction monitoring (PRM) for in vivo target engagement.** Dry peptide samples were reconstituted in water containing 0.1% formic acid (20 µL) and 10 µL were injected onto an EASY-Spray column (15 cm × 75 µm ID, PepMap C18, 3 µm particles, 100 Å pore size, Thermo Fisher Scientific) using a Vanquish Neo UHPLC (Thermo Fisher Scientific). Peptides were separated over a 15 min gradient of 0–40% acetonitrile (0.1% formic acid) and analyzed on an Orbitrap Fusion Lumos (Thermo Fisher Scientific) operated using a parallel reaction monitoring (PRM) method for two AIG1 peptides and additional control peptides to assess sample integrity: FASN, PCCA, PC. Selected ions were isolated and fragmented by high energy collision dissociation (HCD) at 30% CE and fragments were detected in the Orbitrap at 15,000 resolution. Further details for the targeted peptides can be found in Supplementary Table 2.

**MS Acquisition for TMTpro labeled peptides.** MS was performed using a Thermo Scientific Vanquish Neo and Orbitrap Eclipse system. Peptides were eluted over a 240 min nLC 5-25-45% acetonitrile gradient. For all samples, data were collected in data-dependent acquisition mode over a range from 375 to 1500 m/z. Each full scan was followed by fragmentation events for 1.5 sec including real-time search of mouse

peptides with TMTpro modifications and carbamidomethylation of cysteine and oxidation of methionine for subsequent synchronous precursor selection. Dynamic exclusion was enabled (repeat count of 1, exclusion duration of 60 s) for all experiments.

**Reductive dimethylation data analysis.** Data analysis was performed using a custom in-house pipeline. Raw files were converted, searched and processed with the same software as described for the TMTpro datasets, with the following modifications. Comet searches were performed with the following static modifications carbamidomethylation at cysteine [+57.021463 Da], light dimethyl labeling at lysine and the N-terminus [+28.031300 Da], and the following differential modifications: heavy dimethyl labeling at lysine and the N-terminus [+6.031817 Da] and methionine oxidation [+15.994914 Da]. Peptide ratio quantification was then performed using CIMAGE[69]. Competition ratios for each protein were calculated from the median of all peptide ratios, and then normalized so that the median protein ratio for each sample was equal to one. Ratios were capped between -32 and 32 and converted to percent competition values. Percent competition values for serine hydrolases were reported only if the protein had at least 3 quantified peptides, and at least 1 unique quantified peptide, with an exception being made for AIG1, which was included regardless of its numbers of quantified peptides. The pipeline as well as all code, parameters, and FASTA databases used to process the data as described here can be accessed at the following URL: https://doi.org/10.5281/zenodo.10963110.

**TMTpro data analysis.** Data analysis was performed using a custom in-house pipeline. Raw files were converted to the indexed mzML format using ThermoRawFileParser v1.4.3[70], and then searched with Comet v2019.01.5[71] against a mouse FASTA database obtained from UniProt[72] (reference proteome UP000000589_10090, downloaded on 2020-03-21), with one modification: the original sequence for Ddhd1 was swapped with the sequence of the longest Ddhd1 isoform, having accession F8WIJ5, and concatenated with a list of non-mouse contaminant proteins obtained from MaxQuant[73], and reverse decoys. Comet searches were performed with the following static modifications: carbamidomethylation at cysteine [+57.021463 Da], and TMTPro labeling on lysine and the N-terminus [+304.207145 Da], and the following differential modification: oxidation at methionine [+15.994914 Da]. Further processing was performed using the OpenMS platform v3.1.0[74] (commit short hash 26292ad). Files were first converted from.pep.xml to.idXML using IDFileConverter, target/decoy information was added using PeptideIndexer, features were extracted with PSMFeatureExtractor, and Percolator v3.05[75] was run through PercolatorAdapter. Peptide spectrum matches (PSMs) were then filtered with a q-value threshold of 0.01 using IDFilter. MS3 reporter ion intensities and associated noise values were extracted from the most intense peak within a 0.002 Th window of each reporter ion expected mass and the signal-to-noise ratio (SNR) was calculated by dividing the intensity value by the corresponding noise value at that m/z. SNR values were then normalized per channel to the mean per-channel summed SNR. PSMs having a coefficient of variation greater than 0.5 across control channels, or having a mean SNR of less than two across all channels were removed. Protein competition ratios were then calculated by grouping PSMs by protein, calculating the mean control condition SNR across replicate channels, and dividing that figure by the summed SNR for every other channel. Ratios were capped between −32 and 32 and converted to percent competition values. Percent competition values for serine hydrolases were reported only if the protein had at least three quantified peptides, and at least one unique quantified peptide. The pipeline as well as all code, parameters, and FASTA databases used to process the data as described here can be accessed at the following URL: https://doi.org/10.5281/zenodo.10963110.

## Chemistry.

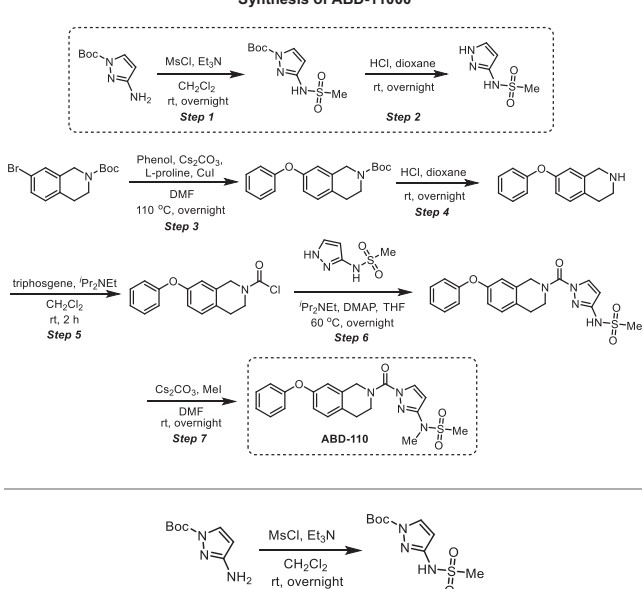

Synthesis of ABD-11000

**Step 1 - *tert*-butyl 3-(methylsulfonamido)-1H-pyrazole-1-carboxylate**: A 100-mL round-bottom flask was charged with tert-butyl 3-amino-1H-pyrazole-1-carboxylate (300 mg, 1.64 mmol, 1.00 equiv), methanesulfonyl chloride (207 mg, 1.81 mmol, 1.10 equiv), dichloromethane (10 mL) and triethylamine (497 mg, 4.91 mmol, 3.00 equiv). The resulting solution was stirred overnight at room temperature and quenched by water (40 mL). The resulting solution was extracted with dichloromethane (3 ×80 mL) and the organic layers were combined, washed with water (3 ×20 mL), dried over anhydrous sodium sulfate, filtered and concentrated under reduced pressure to provide 400 mg (crude) of *tert*-butyl 3-(methylsulfonamido)-1H-pyrazole-1-carboxylate which was carried to the next step without further purification. 1H NMR (400 MHz, CDCl3): δ 8.41 (s, 1H), 8.01 (d, J = 2.9 Hz, 1H), 6.51 (d, J = 2.9 Hz, 1H), 3.11 (s, 3H), 1.66 (s, 9H); MS (ESI, m/z): [M + H]+ calcd. for C9H16N3O4S, 262; found, 262.

**Step 2 - N-(1H-pyrazol-3-yl)methanesulfonamide:** A 100-mL round-bottom flask was charged with *tert*-butyl 3-(methylsulfonamido)-1H-pyrazole-1-carboxylate (400 mg, 1.53 mmol, 1.00 equiv), 1,4-dioxane (10 mL) and concentrated hydrogen chloride (5 mL). The resulting solution was stirred overnight at room temperature and concentrated under reduced pressure to provide 450 mg (crude) of N-(1H-pyrazol-3-yl)methanesulfonamide. 1H NMR (400 MHz, CDCl3): δ 11.67 (s, 2H), 9.78 (s, 1H), 7.64 (d, J = 2.3 Hz, 1H), 6.00 (d, J = 2.3 Hz, 1H), 3.02 (s, 3H); MS (ESI, m/z): [M + H]+ calcd. for C4H8N3O2S, 162; found, 162.

**Step 3 · *tert*-butyl 7-phenoxy-3,4-dihydroisoquinoline-2(1H)-carboxylate:** A 100-mL round-bottom flask was charged with *tert*-butyl 7-bromo-1,2,3,4-tetrahydroisoquinoline-2-carboxylate (2.00 g, 6.41 mmol, 1.00 equiv), *N,N*-dimethylformamide (50 mL), phenol (1.20 g, 12.7 mmol, 2.00 equiv), dicesium carbonate (5.20 g, 16.0 mmol, 2.50 equiv), L-proline (298 mg, 2.57 mmol, 0.40 equiv) and cuprous iodide (244 mg, 1.28 mmol, 0.20 equiv) under nitrogen. The resulting solution was stirred overnight at 110 C and quenched by water (40 mL). The resulting solution was extracted with dichloromethane (3 ×80 mL) and the organic layers were combined, washed with water (3 ×20 mL), dried over anhydrous sodium sulfate, filtered and concentrated under reduced pressure. The residue was chromatographed on a silica gel column with ethyl acetate/petroleum ether (1/10) to provide 1.25 g (60% yield) of *tert*-butyl 7-phenoxy-1,2,3,4-tetrahydroisoquinoline-2-carboxylate as light-yellow oil. 1H NMR (400 MHz, CDCl3): $\delta$ 7.40−7.31 (m, 2H), 7.16−7.09 (m, 2H), 7.05−6.99 (m, 2H), 6.86 (dd, J = 8.3, 2.5 Hz, 1H), 6.78 (d, J = 2.5 Hz, 1H), 4.55 (s, 2H), 3.67 (t, J = 5.9 Hz, 2H), 2.83 (t, J = 5.9 Hz, 2H), 1.51 (s, 9H); MS (ESI, m/z): [M-tBu+H]+ calcd. for $C_{16}H_{16}NO_3$, 270.1; found, 270.0.

**Step 4 · 7-phenoxy-1,2,3,4-tetrahydroisoquinoline:** A 50-mL round-bottom flask was charged with *tert*-butyl 7-phenoxy-1,2,3,4-tetrahydroisoquinoline-2-carboxylate (1.25 g, 3.84 mmol, 1.00 equiv), 1,4-dioxane (10 mL), concentrated hydrogen chloride (5 mL). The resulting solution was stirred overnight at room temperature and concentrated under reduced pressure. The resulting solution was diluted with water (40 mL). The pH value of the solution was adjusted to 11−12 with sodium hydroxide (4 M). The resulting solution was extracted with dichloromethane (3 ×80 mL) and the organic layers were combined, washed with water (3 ×20 mL), dried over anhydrous sodium sulfate, filtered and concentrated under reduced pressure to provide 1.10 g (crude) of 7-phenoxy-1,2,3,4-tetrahydroisoquinoline as light-yellow oil. 1H NMR (400 MHz, CDCl3): $\delta$ 9.58 (s, 2H), 7.47−7.36 (m, 2H), 7.29−7.22 (m, 1H), 7.19−7.11 (m, 1H), 7.04−6.97 (m, 2H), 6.97−6.89 (m, 2H), 4.22 (d, J = 4.5 Hz, 2H), 3.39−3.33 (m, 2H), 2.99 (t, J = 6.3 Hz, 2H); MS (ESI, m/z): [M + H]+ calcd. for $C_{15}H_{16}NO$, 226; found, 226.

**Step 5 · 7-phenoxy-3,4-dihydroisoquinoline-2(1H)-carbonyl chloride:** A 50-mL round-bottom flask was charged with triphosgene (158 mg, 0.530 mmol, 0.50 equiv), dichloromethane (5 mL), 7-phenoxy-1,2,3,4-tetrahydroisoquinoline (240 mg, 1.07 mmol, 1.00 equiv). *N,N*-diisopropylethylamine (413 mg, 3.20 mmol, 3.00 equiv) was added dropwise at 0 °C. The resulting solution was stirred for 2 h at room temperature and quenched with water (40 mL). The resulting solution was extracted with dichloromethane (3 ×80 mL) and the organic layers were combined, washed with water (3 ×20 mL), dried over anhydrous sodium sulfate, filtered and concentrated under reduced pressure to provide 300 mg (crude) of 7-phenoxy-1,2,3,4-tetrahydroisoquinoline-2-carbonyl chloride as light-yellow oil. This was used in the next step without further purification.

**Step 6 · N-(1-(7-phenoxy-1,2,3,4-tetrahydroisoquinoline-2-carbonyl)-1H-pyrazol-3-yl)methanesulfonamide:** A 50-mL round-bottom flask was charged with 7-phenoxy-1,2,3,4-tetrahydroisoquinoline-2-carbonyl chloride (150 mg, 0.520 mmol, 1.00 equiv), tetrahydrofuran (5 mL), *N*-(1H-pyrazol-3-yl)methanesulfonamide (84.0 mg, 0.520 mmol, 1.00 equiv), *N,N*-diisopropylethylamine (202 mg, 1.56 mmol, 3.00 equiv) and 4-dimethylaminopyridine (13.0 mg, 0.110 mmol, 0.20 equiv). The resulting solution was stirred overnight at 60 °C and quenched by water (40 mL). The resulting solution was extracted with dichloromethane (3 ×80 mL) and the organic layers were combined, washed with water (3 ×20 mL), dried over anhydrous sodium sulfate, filtered and concentrated under reduced pressure. The residue was chromatographed on a silica gel column with ethyl acetate/petroleum ether (1/1) to provide 170 mg (79% yield) of *N*-[1-[(7-phenoxy-1,2,3,4-tetrahydroisoquinolin-2-yl)carbonyl]-1H-pyrazol-3-yl] methanesulfonamide as light-yellow oil. 1H NMR (400 MHz, CDCl3): $\delta$ 8.09 (d, J = 2.8 Hz, 1H), 7.39−7.32 (m, 2H), 7.18−7.10 (m, 2H), 7.04− 6.98 (m, 2H), 6.90 (dd, J = 8.3, 2.5 Hz, 1H), 6.78 (d, J = 2.5 Hz, 1H), 6.34 (d, J = 2.9 Hz, 1H), 4.89 (s, 2H), 4.02 (t, J = 5.8 Hz, 2H), 3.16 (s, 3H), 3.01 (t, J = 5.9 Hz, 2H); 13 C NMR (100 MHz, CDCl3): $\delta$ 157.2, 155.8, 150.8, 148.01, 134.1, 133.7, 130.1, 129.8, 129.0, 123.4, 118.8, 117.8, 116.4, 99.7, 40.7, 28.1; HRMS (ESI, m/z): [M + H]+ calcd. for $C_{20}H_{21}N_4O_4S$, 413.1284; found, 413.1284.

**Step 7 · N-methyl-N-(1-(7-phenoxy-1,2,3,4-tetrahydroisoquinoline-2-carbonyl)-1H-pyrazol-3-yl)methanesulfonamide (ABD-110):** A 50-mL round-bottom flask was charged with *N*-[1-[(7-phenoxy-1,2,3,4-tetrahydroisoquinolin-2-yl)carbonyl]-1H-pyrazol-3-yl]methanesulfonamide (170 mg, 0.410 mmol, 1.00 equiv), *N,N*-dimethylformamide (4 mL), dicesium carbonate (269 mg, 0.830 mmol, 2.00 equiv) and iodomethane (88.0 mg, 0.620 mmol, 1.50 equiv). The resulting solution was stirred overnight at room temperature and quenched by water (40 mL). The resulting solution was extracted with dichloromethane (3 ×80 mL) and the organic layers were combined, washed with water (3 ×20 mL), dried over anhydrous sodium sulfate, filtered and concentrated under reduced pressure. The crude product was purified by preparative HPLC using the following gradient conditions: 20% CH3CN/80% Phase A increasing to 80% CH3CN over 10 min, then to 100% CH3CN over 0.1 min, holding at 100% CH3CN for 1.9 min, then reducing to 20% CH3CN over 0.1 min, and holding at 20% for 1.9 min, on a Waters 2767-5 Chromatograph. Column: Xbridge Prep C$_{18}$, 19*150 mm 5um; Mobile phase: Phase A: aqueous NH4HCO3 (0.05%); Phase B: CH3CN; Detector, UV220 and 254 nm. Purification resulted in 103 mg (58% yield) of *N*-methyl-*N*-[1-[(7-phenoxy-1,2,3,4-tetrahydroisoquinolin-2-yl)carbonyl]-1H-pyrazol-3-yl]methanesulfonamide as a light-yellow semi-solid. $^{1}$H NMR (400 MHz, CDCl3) $\delta$ 8.07 (d, J = 2.8 Hz, 1H), 7.31−7.36 (m, 2H), 7.09−7.16 (m, 2H), 6.98−7.00 (m, 2H), 6.87−6.89 (m, 1H), 6.77 (s, 1H), 6.52 (d, J = 2.8 Hz, 1H), 4.89 (s, 2H), 4.03 (s, 2H), 3.39 (s, 3H), 3.02 (*t*, J = 5.8 Hz, 2H), 2.94 (s, 3H); 13 C NMR (100 MHz, CDCl3): $\delta$ 157.2, 155.8, 151.60, 150.9, 134.2, 133.4, 130.1, 129.8, 129.1, 123.4, 118.8, 117.9, 116.4, 101.3, 36.7, 35.6; HRMS (ESI, m/z): [M + H]+ calcd. for $C_{21}H_{23}N_4O_4S$, 427.1440; found, 427.1440.

## Statistics and reproducibility

All data are presented as mean ± s.e.m. Statistical analysis was performed in Prism 8 (GraphPad Software). Comparisons between two groups were performed using an unpaired two-tailed Student's *t* test. A one-way ANOVA followed by Dunnett's or Bonferroni's multiple comparisons test was used to compare more than two groups containing one factor. A *two-way* ANOVA followed by Dunnett's multiple

comparisons test was used to compare more than two groups containing two factors. $P < 0.05$ was considered to be statistically significant.

### Reporting summary

Further information on research design is available in the Nature Portfolio Reporting Summary linked to this article.

## Data availability

All data supporting the findings of this study are available within the paper, in the supplementary information file, and in the source data file. The RNA-seq data generated in this study have been deposited in the NCBI Gene Expression Omnibus and are accessible through accession number GSE213048. The mass spectrometry proteomics data generated in this study are available via ProteomeXchange with identifier PXD047864. All code, including pipeline orchestration, parameters, search databases, and any report-generating scripts used to process the proteomics data in this paper has been open-sourced with a GPL v3.0 license and can be accessed without any restriction at the following URL: https://doi.org/10.5281/zenodo.10963110.

Source data are provided with this paper, including the raw lipidomic data accompanying Figs. 6, 8, and supplementary Figs. 5 and 9.

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

## Acknowledgements

This work was supported by NIH R01 DK102170 to E.D.R. and by support from the Dasman Diabetes Institute/Kuwaiti Ministry of Health to E.D.R. and R.A., NIH grants R01 DK43051 to B.B.K. and R01 DK106210 to B.B.K. and A.Sag, and a grant from the JPB Foundation to B.B.K.; NIH K01 128075 to A. San. We thank the NIH BNORC-supported Functional Genomics and Bioinformatics Core at BIDMC. This work was supported by the Mass Spectrometry Core of the Salk Institute with funding from NIH-NCI CCSG: P30 014195 and the Helmsley Center for Genomic Medicine. The MS data for untargeted lipidomics was gathered on a Thermo Fisher Q Exactive Hybrid Quadrupole Orbitrap mass spectrometer funded by an NIH grant (1S10OD021815-01). We also thank Kerry Wellenstein and Chen Zhenlong for their technical help.

## Author contributions

Conceptualization, S.Y. and E.D.R.; Investigation, S.Y., A.San, N.N., R.M.S., B.R.F., R.A.H., C.L.H., D.M.H., M.J.N., A.M.P.; Data analysis, C.L.J., S.Y.; Writing, S.Y., and E.D.R.; Funding Acquisition, B.B.K., R.A., E.D.R.; Supervision and data interpretation, A. Sag, B.B.K., R.A., and E.D.R.

## Competing interests

The authors declare the following competing interests: M.J.N., N.N., R.M.S., B.R.F., R.A.H., C.L.H., and D.M.H. are employees of Lundbeck Pharmaceuticals. The other authors declare no competing interests.
