## [Peer Review File · Nature Communications]

REVIEWER COMMENTS

Reviewer #1 (Remarks to the Author):

Team of prof. Rosen proposes that chronic inflammation in adipose tissue triggers an interferon regulatory factor 3-dependent mechanism, which promotes the transcription of AIG1 lipid hydrolase. AIG1 reduces levels of FAHFA, lipids produced by adipocytes to improve insulin sensitivity in peripheral tissues. This idea is novel, but it is not the only explanation for adipose tissue meta-inflammation, as is suggested in the text and discussion.

General comments: The manuscript is very hard to read. There are so many links to extended data sets that it is impossible to read the text. I suggest removing these links. Text between lines 127 and 151 is related only to supporting data. I strongly recommend removing/reducing the 'supporting' data and focusing only on the main topic. The path from LTR4/LPS results to identifying the adipokine is very long. The part about the AIG1 role is convincing, but the last part focused on the AIG1 inhibitor needs to be completed.

Major comments:

1. Lipidomics

1a. The identification of OAHFA, Figure 3B

Lines 208-211: "Among several lipid species whose abundance was altered (fold change cutoff ≥ 1.5 , p-value < 0.05) by IRF3 were oleic acid-hydroxy fatty acids (OAHFAs), which are members of the fatty acyl ester of hydroxy fatty acid (FAHFA) family of lipids produced by adipocytes and which have been implicated as important endogenous insulin sensitizers³¹".

There are two OAHFAs in the data table (rows 120 – 121). Both ions are identified as OAHFA(16:0_16:0)–H with m/z 509.4575 and 507.4419, but are referred to as oleic acid-containing lipids in the main text. What is the data for such annotation? This should be clarified.

1b. The volcano plot Figure 3B

How exactly was the volcano plot generated? There are two sheets in the excel file – negative and positive mode. Still, some lipids are present in both polarities (e.g., LPE 16:0). It would be incorrect to include the same lipid (different ion form) in the analysis twice. Please clarify.

The fold change and $-\log(p\text{-val})$ parameters are less strict, and no FDR is applied. Given the huge variability of the dataset (e.g., normalized(?) peak areas for ChE(26:6) +NH₄ range from 1.016941 to 0.002013), better data curation should be performed. Some TG lipids 'expected biological replicates (WTs)' differ ~ 100 times.

The 'OAHFAs' were not significantly altered (t-test, rows 120-121, p-val 0.107115 and 0.153064). Yet, they are highlighted in the volcano plot as lipids with p-val less than 0.05. What about the other lipids that were significantly altered? Why the 'OAHFAs' were selected as the lipids responsible for the effect of F13KO? (and the others were neither mentioned nor explored).

The authors were fortunate to pick the incorrectly annotated, not-significantly changed lipid as their candidate for the non-protein-based adipokine. Please clarify.

1c. I would generally suggest following the Lipidomics Minimal Reporting Checklist (<https://www.nature.com/articles/s42255-022-00628-3>) to consolidate the lipidomics part.

2. Figure 3C and 3D – why are the SVF adipocytes expressed as fold-change and in pmol/mg protein? This could be unified to document the effect. Also, the significant difference in OAHFAs is difficult to believe given the data in extraFig4, panel A. I suppose that the Figure 3C is recalculated sum of isomers shown in extraFig4A, where only isomer 9 is significantly different and when only the relative (not absolute levels of isomers) were averaged (?). This should be clarified.

3. Selection of the AIG1 inhibitor. It is unclear why the authors were looking for the AIG1-only inhibitor. The dual inhibitor would also inhibit the other FAHFA hydrolase. Why was the selectivity profiling performed in the brain while the main effects were shown in adipocytes?

4. Obese mice were injected with the AIG1 inhibitor for two weeks, and the improved insulin sensitivity was documented. The effect on FAHFA concentration is not shown. It is essential to see that the inhibitor reduced the activity of AIG1 and increased/restored the FAHFA levels to confirm the primary result.

5. Line 333. The authors refer to previous AIG1-related results (ref. 33). If I understand correctly, the AIG1 knockout animals are not protected from obesity (ref. 33, long high-fat diet feeding), and the injection of the ABD-110 improved insulin sensitivity of the animals that were already obese (Figure 4). The authors also discuss the role of TG-esterified FAHFAs in ref. 33. These FAHFA-containing lipids are not mentioned in this manuscript but could provide some insight into the cascade regulation.

Minor comments:

1. What is AIG1? The abbreviation for this important gene is explained too late (line 224).

2. Statistics. It needs to be clarified how was some statistics calculated. Fig1C should be 3-way ANOVA (three parameters?), but this is inconsistent with the legend and methods. The same for some other panels.

3. Fold change. Please stay consistent. Extended Data Figure 4A – fold change related to 100. Panel B – fold change to unknown value. Others – fold change relative to 1.

Reviewer #2 (Remarks to the Author):

Summary

The manuscript by Yan et al. explores the long-known link between inflammation and the onset of insulin-resistance and adipose tissue dysfunction in obesity. To this end, the authors demonstrate that the transcription factor IRF3 was required for TLR3/4-induced insulin resistance, i.e. insulin-dependent glucose uptake, in various adipocyte cell models in vitro. While knockdown of IRF3 promoted insulin signaling, overexpression of IRF3 inhibited it. Under thermoneutral conditions, high fat diet (HFD)-fed, adipocyte-specific IRF3 KO mice displayed enhanced insulin sensitivity and improved glucose tolerance with no obvious effects on body weight. In contrast, constitutive overexpression of IRF3 in adipocytes impaired insulin sensitivity with no obvious body weight phenotype under thermoneutral and HFD conditions. Lipidomics analysis showed that the loss of IRF3 in adipocytes led to elevated levels of FAHFA lipid species along with a down-regulation of the FAHFA hydrolase Aig1. Genetic and pharmacological manipulation of Aig1 in adipocytes and adipose tissue restored or reversed the effects of IRF3KO or overexpression on insulin sensitivity, respectively. Overall, the authors conclude that the IRF3/Aig1 axis is an important link between obesity-driven inflammation and insulin resistance in obesity.

General comments

Obesity and its long-term consequences will continue to affect a large proportion of the world's population. As inflammation is an established hallmark of this condition, the current manuscript by Yan et al. clearly addresses a timely and relevant topic in biomedical research. The authors employ a large variety of state-of-the-art technologies to address their experimental questions so that conclusions are generally supported well by the experimental data. The manuscript is well written and concise. The discovery of Aig1 as a downstream IRF3 target and the description of a selective Aig1 inhibitor (ABD-110) represent the main novelties in this report. In contrast, major components of the conceptual framework (IRF3 function in inflammation/adipose tissue; IRF3 as downstream effector of TLR signaling; link between IRF3 and insulin sensitivity; role of FAHFAs in insulin sensitization etc.) are long-known and have been well-established in the literature (including the author's own publications), thereby diminishing the novelty of certain aspects of this study. In addition, two main issues require further attention by the authors: a) The authors should perform lipidomics analysis in their genetic/pharmacological rescue experiments under IRF3 KO or overexpression conditions +/- Aig1 restoration or inhibition. This will be important to clear link the FAHFAs to the proposed IRF3/Aig1 axis under relevant conditions. b) The development of an Aig1-selective inhibitor is potentially very interesting and novel. Thus, the authors should provide more data on the efficacy and selectivity of this new compound. How does ABD-110 work in wild-type mice under chow conditions? Does it show effects in IRF3 KO animals? Please demonstrate target engagement in the HFD experiment. Do you observe a dose response in terms of insulin sensitization? Does ABD-110 also work in human adipocytes? What is

the plasma half-life and tissue distribution after daily injections? Addition of corresponding data will significantly strengthen the case for publication.

Specific comments

1. Extended Figure 7: Please provide Aig1 WG data on liver and skeletal muscle. How are the effects of eWAT-selective Aig1 reconstitution on liver and skeletal muscle p-Akt explained? By directly or indirectly modulating FAHFA levels? Please comment.

Reviewer #3 (Remarks to the Author):

In this manuscript, Shuai Yan and coworkers provide experimental evidence that interferon regulatory factor 3 (IRF3) mediates impaired glucose homeostasis via activation of the endogenous FAHFA hydrolase AIG1 in adipocytes. The authors use an adipocyte-specific IRF3 knockout model and demonstrate that mice with a deletion of IRF3 are (partly) protected against high-fat diet-induced insulin resistance. In contrast, adipocyte IRF3 overexpression is associated with impaired insulin sensitivity. Pharmacological inhibition of AIG1 was sufficient to reverse insulin resistance and to improve glucose metabolism parameters in adipocyte specific IRF3 overexpressors. The authors conclude that the IRF3/AIG1 axis plays an important role in adipocytes as mediator of obesity-related inflammation and insulin resistance.

The manuscript is timely, very well written and adds important data to our understanding how "immunometabolism" mechanisms are regulated at the level of adipocytes by IRF3-related mechanisms. The applied model systems are state-of-the art and the well controlled, extensive experiments support the conclusions drawn. However, there are a few points that should be clarified or addressed:

1) Although briefly mentioned in the introduction, the selection of IRF3 among different (potentially equally important) mediators of obesity-related metaflammation as primary target of investigation should be justified in a little more detail. In addition to IRFs and NF- κ B, are there other candidate pathways?

2) Are histology example slides from the transgenic mouse models (ref. 25) available to reiterate effects of IRF3 on adipocyte size and immune cell infiltration?

3) Are there effects of IRF3 knockdown and/or overexpression on insulin sensitivity as assessed through lipolysis inhibition?

Minor:

- 1) IRF3 should be spelled out for the first time in the title and abstract.
- 2) People first language should be used to describe people with obesity.

Response to the Reviewers

We thank the reviewers for their thoughtful comments. We have now performed additional experiments that address their concerns. The comments of the reviewers are in **bold**, and our responses are in plain text.

Reviewer #1

General comments: The manuscript is very hard to read. There are so many links to extended data sets that it is impossible to read the text. I suggest removing these links. Text between lines 127 and 151 is related only to supporting data. I strongly recommend removing/reducing the 'supporting' data and focusing only on the main topic. The path from LTR4/LPS results to identifying the adipokine is very long. The part about the AIG1 role is convincing, but the last part focused on the AIG1 inhibitor needs to be completed.

I believe this is the first time we have ever been told to include *less* data, which is refreshing. Respectfully, however, we feel strongly that the supporting data are critical to inform the reader of what we did and why we did it. To make the manuscript easier to read, we have altered the text, changed the figure number and order, and consolidated the callouts—hopefully this new version will be more straightforward to follow.

Major comments:

1. Lipidomics

1a. The identification of OAHFA, Figure 3B

Lines 208-211: “Among several lipid species whose abundance was altered (fold change cutoff ≥ 1.5 , p-value < 0.05) by IRF3 were oleic acid-hydroxy fatty acids (OAHFAs), which are members of the fatty acyl ester of hydroxy fatty acid (FAHFA) family of lipids produced by adipocytes and which have been implicated as important endogenous insulin sensitizers³¹”.

There are two OAHFAs in the data table (rows 120 – 121). Both ions are identified as OAHFA (16:0_16:0) –H with m/z 509.4575 and 507.4419, but are referred to as oleic acid-containing lipids in the main text. What is the data for such annotation? This should be clarified.

1b. The volcano plot Figure 3B

How exactly was the volcano plot generated? There are two sheets in the excel file – negative and positive mode. Still, some lipids are present in both polarities (e.g., LPE 16:0). It would be incorrect to include the same lipid (different ion form) in the analysis twice. Please clarify.

The fold change and $-\log$ (p-val) parameters are less strict, and no FDR is applied. Given the huge variability of the dataset (e.g., normalized(?) peak areas for ChE(26:6) +NH4 range from 1.016941 to 0.002013), better data curation should be performed. Some TG lipids 'expected biological replicates (WTs)' differ ~100 times.

The ‘OAHFAs’ were not significantly altered (t-test, rows 120-121, p-val 0.107115 and 0.153064). Yet, they are highlighted in the volcano plot as lipids with p-val less than 0.05. What about the other lipids that were significantly altered? Why the ‘OAHFAs’ were selected as the lipids responsible for the effect of FI3KO? (and the others were neither mentioned nor explored).

The authors were fortunate to pick the incorrectly annotated, not-significantly changed lipid as their candidate for the non-protein-based adipokine. Please clarify.

1c. I would generally suggest following the Lipidomics Minimal Reporting Checklist to consolidate the lipidomics part.

The reviewer correctly points out that our untargeted lipidomics dataset had a large amount of variability, and that the increase in the OAHFAs (in the untargeted set) was not statistically significant. This is not entirely surprising, given that FAHFA species are found in low abundance. Accordingly, we have removed the untargeted lipidomics completely from the manuscript. The discovery that AIG1 was altered provides sufficient justification to look at FAHFAs using targeted lipidomics (where results were highly significant). In addition, while we agree with the reviewer that there are other interesting lipid species that change with

IRF3 KO, this manuscript is about the link between IRF3, AIG1, and FAHFAs. We will follow up some of these other lipid species in later work.

2. Figure 3C and 3D – why are the SVF adipocytes expressed as fold-change and in pmol/mg protein? This could be unified to document the effect.

Thanks for catching the discrepancy—we have now unified the way the data are presented.

Also, the significant difference in OAHSAs is difficult to believe given the data in extraFig4, panel A. I suppose that the Figure 3C is recalculated sum of isomers shown in extraFig4A, where only isomer 9 is significantly different and when only the relative (not absolute levels of isomers) were averaged (?). This should be clarified.

Thanks for pointing this out. We have now repeated the FAHFA measurement experiment using a targeted lipidomic method which was developed by the Saghatelian and Kahn labs (PMID: 26985573) because FAHFAs are low in abundance. This method has much more sensitivity and precision for FAHFAs compared to untargeted lipidomics. In Fig. 6a and Supplementary Fig. S5a, we now show that FAHFA levels are increased in FI3KO cells.

3. Selection of the AIG1 inhibitor. It is unclear why the authors were looking for the AIG1-only inhibitor. The dual inhibitor would also inhibit the other FAHFA hydrolase.

As the reviewer points out, a dual inhibitor would also block ADTRP, and would therefore not inform us about the specific role of AIG1. While inhibiting ADTRP might tell us something about the utility of inhibiting all FAHFA degradation as an insulin-sensitizing strategy, our studies show that IRF3 induces AIG1 (and not ADTRP), and so the specific inhibitor provides more precise mechanistic insight into our question.

Why was the selectivity profiling performed in the brain while the main effects were shown in adipocytes?

AIG1 is a small protein (27 kDa) with multiple transmembrane domains leading to few proteotypic peptides that can be detected and quantified in MS-based proteomics experiments. Because of this, AIG1 needs to be highly expressed for detection in MS-based proteomics experiments. The brain expresses a high amount of AIG1 as well as a diverse set of serine hydrolases thus providing an excellent matrix for broadly characterizing AIG1 potency and serine hydrolase selectivity.

We agree that selectivity profiling in matrices more relevant to the scope of this paper is warranted here. To address this, we have added MS-based ABPP selectivity profiles for ABD-110 in kidney and WAT proteomes. These matrices extend the selectivity panel of enzymes and further support the claim that ABD-110 is highly selective for AIG1. Unfortunately, for the reasons mentioned above, we were unable to detect AIG1 in WAT. Nevertheless, this profile adds confidence that ABD-110 has few off-targets across serine hydrolases expressed in adipose tissue.

4. Obese mice were injected with the AIG1 inhibitor for two weeks, and the improved insulin sensitivity was documented. The effect on FAHFA concentration is not shown. It is essential to see that the inhibitor reduced the activity of AIG1 and increased/restored the FAHFA levels to confirm the primary result.

In Fig. 8f, g and Supplemental Fig. 9c we now show that the inhibitor increases FAHFA levels as predicted.

5. Line 333. The authors refer to previous AIG1-related results (ref. 33). If I understand correctly, the AIG1 knockout animals are not protected from obesity (ref. 33, long high-fat diet feeding), and the

injection of the ABD-110 improved insulin sensitivity of the animals that were already obese (Figure 4). The authors also discuss the role of TG-esterified FAHFAs in ref. 33. These FAHFA-containing lipids are not mentioned in this manuscript but could provide some insight into the cascade regulation.

We have noted an increase in high carbon length TGs in adipocytes lacking IRF3—these species could include FAHFAs esterified into TGs, which would be consistent with our current story (i.e., if IRF3 is not able to induce AIG1, free FAHFA levels climb, and esterification may be how the cell deals with this). At this time, however, we don't have data in hand to prove that the levels of TG esterified FAHFAs is elevated, so we have not mentioned this possibility.

Minor comments:

1. What is AIG1? The abbreviation for this important gene is explained too late (line 224).

We now define AIG1 in the abstract.

2. Statistics. It needs to be clarified how was some statistics calculated. Fig1C should be 3-way ANOVA (three parameters?), but this is inconsistent with the legend and methods. The same for some other panels.

This has been fixed.

3. Fold change. Please stay consistent. Extended Data Figure 4A – fold change related to 100. Panel B – fold change to unknown value. Others – fold change relative to 1.

This has been fixed.

Reviewer #2 (Remarks to the Author):

General comments

Obesity and its long-term consequences will continue to affect a large proportion of the world's population. As inflammation is an established hallmark of this condition, the current manuscript by Yan et al. clearly addresses a timely and relevant topic in biomedical research. The authors employ a large variety of state-of-the-art technologies to address their experimental questions so that conclusions are generally supported well by the experimental data. The manuscript is well written and concise.

We thank the reviewer for acknowledging the timeliness and importance of these studies.

The discovery of Aig1 as a downstream IRF3 target and the description of a selective Aig1 inhibitor (ABD-110) represent the main novelties in this report. In contrast, major components of the conceptual framework (IRF3 function in inflammation/adipose tissue; IRF3 as downstream effector of TLR signaling; link between IRF3 and insulin sensitivity; role of FAHFAs in insulin sensitization etc.) are long-known and have been well-established in the literature (including the author's own publications), thereby diminishing the novelty of certain aspects of this study.

It is true that we have published that IRF3 promotes insulin resistance, although that work was mostly performed in global knockout mice which also have altered body weight due to changes in energy expenditure. The novelty of the current work lies in (a) demonstrating that the adipocyte is a key cell type mediating the effect of IRF3 on insulin action, (b) showing definitively that the effect on insulin sensitivity is

independent of weight changes, and most importantly, (c) identifying a specific molecular link between IRF3 and insulin resistance (i.e., reduction of FAHFAs by promoting expression of a FAHFA hydrolase). These are all novel findings. In addition, we introduce an entirely novel compound, ABD-110, which has value as a tool compound, as well as potential therapeutic utility.

In addition, two main issues require further attention by the authors:

a) The authors should perform lipidomics analysis in their genetic/pharmacological rescue experiments under IRF3 KO or overexpression conditions +/- Aig1 restoration or inhibition. This will be important to clear link the FAHFAs to the proposed IRF3/Aig1 axis under relevant conditions.

This issue was brought up Reviewer 1 as well. We have now performed targeted lipidomics in the conditions mentioned by the reviewer, and we see the predicted changes in FAHFA levels. Please see Fig. 8f, g and Supplemental Fig. 9c.

b) The development of an Aig1-selective inhibitor is potentially very interesting and novel. Thus, the authors should provide more data on the efficacy and selectivity of this new compound. How does ABD-110 work in wild-type mice under chow conditions? Does it show effects in IRF3 KO animals? Please demonstrate target engagement in the HFD experiment. Do you observe a dose response in terms of insulin sensitization? Does ABD-110 also work in human adipocytes? What is the plasma half-life and tissue distribution after daily injections? Addition of corresponding data will significantly strengthen the case for publication.

We have now added the following data related to the novel AIG1-selective inhibitor:

1. Accurate mAIG1 IC50s in triplicate for ABD-110 using gel-based ABPP (Supplementary Fig. 7b and c)
2. In vitro confirmation that ABD-110 does not inhibit mADTRP (Supplementary Fig. 7d)
3. In depth MS-ABPP selectivity profiles in mouse brain (Supplementary Fig. 7e), kidney (Supplementary Fig. 7f) and WAT proteomes (Supplementary Fig. 7g)
 - The primary off-targets are CESs and ABHD6 (these targets are very frequently inhibited by carbamates such as ABD-110). Overall, this is a very clean selectivity profile for a serine hydrolase inhibitor and we don't anticipate the few off-targets observed interfering with interpretation of the data presented herein.
4. For covalent inhibitors, the pharmacodynamic effect is generally uncoupled from compound levels. Since ABD-110 is a covalent, irreversible inhibitor of AIG1, we feel that target engagement is a superior metric for compound efficacy than compound half-life and exposure. Accordingly, we have measured in vivo target engagement on AIG1 in iWAT, eWAT, brain and kidney from mice on both chow and HFD (Supplementary Fig. 7i and j)
5. Effect of ABD-110 in chow-fed animals in Supplemental Fig. 8. As expected, we don't see an effect on GTT, ITT, glucose, or fasting insulin (a-c). ABD-110 treatment ameliorates HFD-induced insulin resistance, glucose intolerance, and hyperinsulinemia in WT HFD-feeding mice (d-f).

Specific comments

1. Extended Figure 7: Please provide Aig1 WG data on liver and skeletal muscle. How are the effects of eWAT-selective Aig1 reconstitution on liver and skeletal muscle p-Akt explained? By directly or indirectly modulating FAHFA levels? Please comment.

We provide Aig1 WB data on liver and skeletal muscle (Fig. 7f). AIG1 overexpression doesn't affect AIG1 levels in both liver and skeletal muscle.

Yes, our data indicate that a soluble factor or factors from FI3KO adipocytes causes insulin sensitization in other tissues (Fig. 5a). AIG1 reconstitution reduces this, and our data indicate that it does so by reducing

FAHFA levels.

Reviewer #3 (Remarks to the Author):

The manuscript is timely, very well written and adds important data to our understanding how "immunometabolism" mechanisms are regulated at the level of adipocytes by IRF3-related mechanisms. The applied model systems are state-of-the art and the well controlled, extensive experiments support the conclusions drawn.

We thank the reviewer for these positive comments.

However, there are a few points that should be clarified or addressed:

1) Although briefly mentioned in the introduction, the selection of IRF3 among different (potentially equally important) mediators of obesity-related metaflammation as primary target of investigation should be justified in a little more detail. In addition to IRFs and NF- κ B, are there other candidate pathways?

There are, of course, other pro-inflammatory transcription factors, such as AP-1 proteins, and they may also mediate some of the insulin resistance associated with "metainflammation". We make no claim that the pathway we identify (IRF3 \rightarrow AIG1 \rightarrow reduced FAHFA levels) is the only one that matters. This manuscript, however, is focused on this pathway, inspired by our prior work suggesting an important role for IRF3. We have concerns that broadening the focus to other factors and pathways risks turning this into a review article and not a discrete piece of original investigation. Having said that, we now mention in the Discussion that other pathways may also be involved.

2) Are histology example slides from the transgenic mouse models (ref. 25) available to reiterate effects of IRF3 on adipocyte size and immune cell infiltration?

As pointed out by the reviewer, this topic was covered extensively in our published work. This manuscript strikes off from those data in a new direction. We are leery of republishing data from prior work in the current paper.

3) Are there effects of IRF3 knockdown and/or overexpression on insulin sensitivity as assessed through lipolysis inhibition?

We now provide new data showing that insulin is better able to repress lipolysis in FI3KO adipocytes (Supplemental Fig. 1c). Conversely, insulin is less able to repress lipolysis in FI3OE cells (Supplemental Fig. 2c). These data are consistent with the effects of IRF3 on insulin-stimulated pAKT and glucose uptake.

Minor:

1) IRF3 should be spelled out for the first time in the title and abstract.

We have made this change.

2) People first language should be used to describe people with obesity.

We have made this change.

REVIEWER COMMENTS

Reviewer #1 (Remarks to the Author):

The authors responded to my comments and improved the manuscript. However, I still miss the critical part linking the Aig1 effect in adipocytes and the insulin-sensitizing effect in target organs (changes in circulating FAHFA levels as the putative mediators of the inhibitor effect).

Re 1. Lipidomics.

I understand the decision of the authors to remove the lipidomics part. However, this part of the story now looks like a miracle. Line 296 - "One significantly affected gene was androgen-induced gene 1 (Aig1)". Looking at the "WT VS Tg" supplementary table, I have no idea why the authors chose Aig1 - sorted by FDR, Aig1 is the 81st gene from the top. The lipidomics part at least gave some clues as to why Aig1 was chosen for further study. There is no scientific reason to pick Aig1 and not explore the more affected genes. This is wrong.

Re 2. Targeted lipidomics.

Based on the data in "Supplementary Figure 5", the reduction/increase in FAHFA levels is not consistent with the Aig1 hydrolase preferences published by Parsons et al. Nat Chem Biology 2016. The hydrolysis rate of FAHFAs should be much higher for FAHFAs with branching distal to the carboxylate head group of the lipid. This does not support a specific role for Aig1. Also relevant to comment #5: The TG-esterified FAHFAs discovered by the co-authors could be involved. This part is weakly supported by the data.

Re 4. FAHFA levels.

This question is not answered. The essential question is whether FAHFA levels in blood (serum or plasma) were affected by the inhibitor. Please provide circulating FAHFA levels that could be related to 2DG uptake effects in target organs.

This is also relevant to line 294 "Levels of insulin-sensitizing metabolites or lipids". If the FAHFAs act as insulin-sensitizing lipids, the altered levels in the circulation (reaching the target organs) need to be shown.

One minor comment: The 62MB Excel supplemental file is a nonsense. The Excel file should not be used to store such image data. Please use standardized formats for data exchange.

Reviewer #2 (Remarks to the Author):

The authors have added a significant amount of relevant data to address the referee's comments. Thereby, the manuscript has improved substantially.

Reviewer #3 (Remarks to the Author):

The authors successfully addressed all my comments. I do not have any additional suggestions.

Response to the Reviewer

Reviewer #1

The authors responded to my comments and improved the manuscript. However, I still miss the critical part linking the Aig1 effect in adipocytes and the insulin-sensitizing effect in target organs (changes in circulating FAHFA levels as the putative mediators of the inhibitor effect).

Re 1. Lipidomics.

I understand the decision of the authors to remove the lipidomics part. However, this part of the story now looks like a miracle. Line 296 - "One significantly affected gene was androgen-induced gene 1 (Aig1)". Looking at the "WT VS Tg" supplementary table, I have no idea why the authors chose Aig1 - sorted by FDR, Aig1 is the 81st gene from the top. The lipidomics part at least gave some clues as to why Aig1 was chosen for further study. There is no scientific reason to pick Aig1 and not explore the more affected genes. This is wrong.

Respectfully, it is neither "wrong", nor is it a "miracle". It was just old-fashioned hard work. We knew that we were looking for a non-proteinaceous secreted product that could sensitize cells to insulin. There are a few such compounds known, including FAHFAs, which were discovered by our co-author Barbara Kahn. When we got the transcriptomic data, we went through the list of altered genes meticulously, performing dozens of literature searches until we noted that AIG1 was affected, and was also a FAHFA hydrolase. We were extremely fortunate to have Dr. Kahn and her group as next-door neighbors, and so we enlisted them to help with the targeted lipidomics. This is the true story of how the IRF3→AIG1→FAHFA axis was identified.

Re 2. Targeted lipidomics.

Based on the data in "Supplementary Figure 5", the reduction/increase in FAHFA levels is not consistent with the Aig1 hydrolase preferences published by Parsons et al. Nat Chem Biology 2016. The hydrolysis rate of FAHFAs should be much higher for FAHFAs with branching distal to the carboxylate head group of the lipid. This does not support a specific role for Aig1.

The reviewer is presumably referring to Fig. 3 of the Parsons paper, which shows hydrolysis rates for different FAHFA isomers in HEK293 cells transfected with an AIG1 expression plasmid. In our Supplementary Figure 5, we are not measuring hydrolysis rate—instead, we report steady-state levels of FAHFAs. This distinction is important for several reasons. First, the hydrolysis rate is assessed by adding a standard amount of purified FAHFAs to AIG1-transfected cells. In our data, we are looking at endogenous levels of FAHFAs, for which concentrations among isomers vary by >10-fold. Moreover, in the Parsons paper, only one FAHFA is added at a time; thus, any effect

of competition for AIG-1 binding is eliminated. That is not the case in our experiments, where all FAHFA families are present at natural concentrations and presumably competing with one another for target site occupancy. Taken together, we see no inherent discrepancy between the Parsons data and what we present in Supplementary Figure 5.

Also relevant to comment #5: The TG-esterified FAHFAs discovered by the co-authors could be involved. This part is weakly supported by the data.

We are not sure what the reviewer is asking or suggesting here. In our prior response, we mentioned that elevated free FAHFA levels in the setting of AIG1 inhibition or genetic repression could lead to an increase in TG-esterified FAHFAs. We agree this idea is weakly supported by the data: in fact, we wrote: "At this time, however, we don't have data in hand to prove that the levels of TG esterified FAHFAs are elevated, so we have not mentioned this possibility." It is also possible that increased hydrolysis of the ester bond linking FAHFAs to glycerol as acyl chains in FAHFA-TGs could result in elevated non-esterified FAHFA levels. Digging into this, however, would not strengthen the conclusions of our manuscript.

Re 4. FAHFA levels.

This question is not answered. The essential question is whether FAHFA levels in blood (serum or plasma) were affected by the inhibitor. Please provide circulating FAHFA levels that could be related to 2DG uptake effects in target organs. This is also relevant to line 294 "Levels of insulin-sensitizing metabolites or lipids". If the FAHFAs act as insulin-sensitizing lipids, the altered levels in the circulation (reaching the target organs) need to be shown.

We understand the reviewer's reasoning here, but we don't fully agree. The reviewer is making the assumption that the FAHFAs are being released by the adipose tissue and then traveling to the liver and muscle to cause insulin sensitization. This is sensible, but it's not the only possible model. For example, adipose-derived FAHFAs cause insulin sensitization directly in the adipose tissue, which reduces serum insulin levels in mice on a high fat diet. This reduction in hyperinsulinemia then improves insulin action distally. Similarly, local FAHFAs, which are known to have potent anti-inflammatory effects, may reduce cytokine production in WAT, which can mediate systemic insulin sensitization. Other scenarios are also possible. Moreover, the interpretation of FAHFA levels in serum can be complicated; as an example, adipose-specific knockout of ATGL, a FAHFA biosynthetic enzyme, lowers adipose and serum levels of FAHFAs suggesting that adipose is an important source of serum levels. In contrast, FAHFAs are up-regulated in WAT during fasting, but reduced in serum in the same mice under the same conditions (Yore M and Syed I et al., 2014). Most likely, multiple factors contribute to the regulation of bioavailable FAHFA in the serum, which may be part of the problem.

Nevertheless, at the reviewers' request, we have measured FAHFA levels in the serum of FI3OE and FI3KO mice (see Reviewer Only Fig. 1). The data show that FI3KO mice indeed exhibit significantly elevated total POHSAs, PAHSAs, and OAHSAAs, as the reviewer predicted. The situation is less clear for FI3OE mice, in which there is no real change in total FAHFA levels (oddly, we see a significant elevation in total POHSAs, but since these exist at 10-fold lower concentrations than PAHSAs or OAHSAAs, the overall serum FAHFA level is not changed in the

F13OE mice). As described above, we don't view concordant changes in serum FAHFAs as critical for our conclusions to be sound. Accordingly, we have not included these data in the manuscript, but we can do so if the reviewer or editor deem it necessary.

To make our thoughts on this clearer to readers, we have added a section to the discussion (see lines 411-425).

Reviewer Only Fig. 1. Total POHSAs, PAHSAs, and OAHSAs from serum of F13OE (**top**) and F13KO (**bottom**) mice vs. WT controls. Male mice were killed in the ad lib fed state. *p<0.05

One minor comment: The 62MB Excel supplemental file is a nonsense. The Excel file should not be used to store such image data. Please use standardized formats for data exchange.

Nature Communications explicitly asks for Source data to be sent as an Excel file. In their Instructions to Authors, they write: *For relevant manuscripts, we may request a source data file in Microsoft Excel format or a zipped folder. The source data file should, as a minimum, contain the raw data underlying any graphs and charts, and uncropped versions of any gels or blots presented in the figures. Within the source data file, each figure or table (in the main manuscript and in the Supplementary Information) containing relevant data should be represented by a single sheet in an Excel document, or a single .txt file or other file type in a zipped folder. Blot and gel images should be pasted in and labelled with the relevant panel and identifying information such as the antibody used. We also encourage authors to include any other types of raw data that may be appropriate. An example source data file is available demonstrating the correct format.*

REVIEWERS' COMMENTS

Reviewer #1 (Remarks to the Author):

The authors responded to my comments and improved the manuscript.

RE 1. Now I understand the reason. Thank you for the explanation.

RE 2. Thank you for the explanation.

RE 3. The new lines 411-425 help to understand the limitations. Please include the Reviewer Only Fig. 1 as one of the supplementary figures linked to lines 411-425. This is really important information discussing the systemic and local changes.